# CD81+ senescent-like fibroblasts exaggerate inflammation and activate neutrophils via C3/C3aR1 axis in periodontitis

Liangliang Fu[†], Chenghu Yin[†], Qin Zhao, Shuling Guo, Wenjun Shao, Ting Xia, Quan Sun, Liangwen Chen, Jinghan Li, Min Wang*, Haibin Xia*

State Key Laboratory of Oral and Maxillofacial Reconstruction and Regeneration, Key Laboratory of Oral Biomedicine Ministry of Education, Hubei Key Laboratory of Stomatology, School and Hospital of Stomatology, Wuhan University, Wuhan, China

*For correspondence:
83wangmin@whu.edu.cn (MW);
xhaibin@whu.edu.cn (HX)

[†]These authors contributed equally to this work

Competing interest: The authors declare that no competing interests exist.

## eLife Assessment

This **valuable** study identifies a population of CD81-positive fibroblasts showing senescence signatures that can activate neutrophils through the C3/C3aR1 axis, hence contributing to the inflammatory response in periodontitis. **Solid** evidence, combining in vitro and in vivo analyses and mouse and human data, supports these findings. The revised manuscript has addressed many concerns significantly. The work would be of interest to researchers working in the senescence and oral medicine fields.

**Abstract** Periodontitis, a prevalent inflammatory disease worldwide, poses a significant economic burden on society and the country. Previous research has established a connection between cellular senescence and periodontitis. However, the role and mechanism of cell senescence in the progression of periodontitis have not been thoroughly investigated. This study aimed to explore the involvement of cellular senescence in the pathogenesis of periodontitis and determine the underlying mechanisms. Our findings demonstrated that senescent cells accumulated during the progress of periodontitis in both human samples and mice models. Moreover, several scRNA-seq analyses suggested that gingival fibroblasts were the main cell population undergoing cellular senescence during human periodontitis, which helps mitigate tissue damage and bone loss. Furthermore, we identified a high expression of CD81 in the senescent gingival fibroblast population. These cells were found to actively contribute to inflammation through their potent pro-inflammatory metabolic activity and secretion of senescence-associated secretory phenotype factors. Additionally, they recruited neutrophils via the C3/C3aR1 pathway, indirectly sustaining the inflammatory response. Senolytics via Navitoclax successfully alleviated inflammation and bone loss in periodontitis, and administration of metformin could alleviate inflammation and bone loss in periodontitis through inhibiting cellular senescence. These results provide valuable insights into the cellular and molecular basis of periodontitis-induced tissue damage, highlighting the significance of fibroblast senescence. In conclusion, our study sheds light on the relationship between CD81 and cellular senescence, suggesting its potential as a therapeutic target for periodontitis.

## Introduction

Periodontitis is an inflammatory disease of irreversible progressive tissue damage, alveolar bone loss, and destruction of tooth supporting tissues and is caused by microbial infections that eventually lead to tooth loosening and eventual tooth loss (*Wolff et al., 1994*). Periodontitis affects 11.2% of the global population and more than 40% of people over the age of 30, posing a major burden on public health (*Eke et al., 2020*; *Sanz et al., 2020*). Clinical studies have shown that the prevalence and severity of periodontitis increase with age, and moderate loss of alveolar bone and periodontal attachment is common in older adults (*Huttner et al., 2009*).

Cell senescence is a stress response characterized by irreversible proliferation arrest, resistance to apoptosis, and secretion of a range of inflammatory cytokines, growth factors, and proteases, known as senescence-associated secretory phenotypes (SASPs) (*Coppé et al., 2010*; *Rodier et al., 2009*). Cellular senescence is considered necessary for tissue homeostasis as it aims to eliminate unnecessary damage and promote tissue repair through immune-mediated mechanisms and even prevent the occurrence of tumors (*Campisi, 2013*; *Ohtani and Hara, 2013*). However, the specific environment of the gingival sulcus leads to persistent plaque in periodontal tissue, resulting in oxidative DNA damage as collateral damage of chronic bacterial infection (*Aquino-Martinez et al., 2020*). Repeated exposure to lipopolysaccharide (LPS) derived from *Porphyromonas gingivalis* (Pg), a key pathogen of periodontitis, can accelerate cellular senescence driven by DNA damage (*Aquino-Martinez et al., 2021*). Furthermore, recent evidence suggests that bacteria can also induce senescence of healthy cells in an active oxygen-dependent manner by causing inflammation and excessive neutrophil activity (*Guo et al., 2024*; *Lagnado et al., 2021*). The aggravation or persistence of these stimulating factors can lead to abnormal accumulation of senescent cells and directly affect periodontal tissue function. Therefore, chronic bacterial infections can cause cell senescence through both direct and indirect mechanisms.

Senescent cells have been found to contribute to bacteria-induced inflammation, with the activation of SASPs playing a crucial role in the release of various pro-inflammatory factors, including interleukin (IL)-1α, IL-6, and IL-8. Elevated levels of these inflammatory factors have been associated with periodontal damage and loss of alveolar bone (*Aquino-Martinez et al., 2020*; *Yu et al., 2024*). However, the specific mechanism by which senescent cells contribute to the development of periodontitis remains unclear. In the immune response to periodontitis, dendritic cells infected by Pg activate related SASPs, such as IL-1β, IL-6, and IL-8, which ultimately accelerate the progression of periodontitis (*El-Awady et al., 2022*). Additionally, the aging of T lymphocytes, which are crucial for adaptive immunity, leads to a significant alteration in their immunosuppressive ability in Th17/Treg subsets. This alteration ultimately results in the loss of tooth support and alveolar bone (*González-Osuna et al., 2022*). However, the role and mechanism of cellular senescence in the progression of periodontitis have not been thoroughly investigated.

The breakthrough technology of single-cell RNA sequencing has made it easier to analyze gene expression at the cellular level and identify key cell subpopulations (*Zhang et al., 2021*). In this study, we utilized bulk RNA-seq, clinical periodontal samples, and a mice ligature-induced periodontitis (LIP) model to demonstrate that cellular senescence levels increase with periodontitis progression. Through scRNA-seq, in vitro, and in vivo experiments, we observed significant cellular senescence in gingival fibroblasts. Additionally, we identified a unique subgroup of gingival fibroblasts with high expression of CD81, which exhibited senescence characteristics such as ROS accumulation and enrichment of senescence genes. We propose that this subgroup of fibroblasts can directly promote the progression of periodontitis by secreting SASP-related factors, such as IL-6, and indirectly amplify inflammation by recruiting neutrophils through the complement pathway, specifically C3. We also found that targeting cellular senescence with senolytic drug or metformin can reduce inflammation and delay alveolar bone resorption in periodontitis.

## Results

### Cellular senescence characteristics in periodontitis

Cellular senescence is a manifestation of aging at the cellular level. Although accumulation of senescent cells is normal in aged tissues, persistent bacterial infection and chronic inflammation promote the early onset of senescence by ROS activation and DNA damage (*Aquino-Martinez, 2023*). In

clinical gingival specimens from periodontally healthy individuals of similar age and those diagnosed with periodontitis, we found that the senescence biomarker senescence-associated β-galactosidase (SA-β-gal) was scarcely expressed in the gingiva of young healthy individuals. However, in gingival samples from patients with periodontitis, a notable increase in SA-β-gal-positive cells was observed, primarily localized in the lamina propria of gingival connective tissue (*Figure 1A*). Additionally, immunohistochemical (IHC) staining analysis revealed that other senescent biomarkers, such as cell cycle inhibitory proteins p16 and p21, and senescence-associated heterochromatin foci like H3K9me3, were significantly upregulated in human periodontitis gingival tissues as well (*Figure 1B*).

The clinical samples from periodontitis patients were often derived from older individuals, because periodontitis incidence obviously increases with age (*Eke et al., 2020*). To avoid confounding factors like age potentially affecting the experimental results, we also examined the levels of cellular senescence in the ligature-induced periodontitis (LIP) mouse model (*Figure 1C*). IHC staining results indicated that the protein expression level of p16 among gingiva was significantly upregulated following ligation, peaking at day 7 post-ligation (*Figure 1D*). And then, gingiva at day 7 post-ligation and healthy gingiva as control were collected for protein and gene analysis. Western blotting analysis showed that the protein levels of p16 in the LIP 7D group were about two times larger than those in the control group (*Figure 1E*). And the transcription of *Cdkn2a* (p16 encoding gene), *Cdkn1a* (p21 encoding gene), and *Trp53* (p53 encoding gene) in gingival tissues was higher at day 7 post-ligation than those in control (*Figure 1F*). Furthermore, bulk RNA sequencing was performed on gingival tissues from LIP 7D and healthy mice (*Figure 1—figure supplement 1A*), identifying 458 upregulated and 358 downregulated genes. Notably, among the upregulated genes, 19 senescence-associated genes were detected, including *C3*, *Il-6*, and so on (*Figure 1—figure supplement 1B*). We also observed a significant upregulation of several SASP genes such as *Icam1*, *Mmp3*, *Nos3*, *Igfbp7*, *Igfbp4*, *Mmp14*, *Timp1*, *Ngf*, *Il-6*, *Areg*, and *Vegfa* in the LIP group (*Figure 1—figure supplement 2A*). The gene set enrichment analysis (GSEA) based on our sequencing data revealed the upregulation of the cellular senescence pathway in LIP mice (*Figure 1—figure supplement 1C*). Moreover, a significant reduction in oxidative phosphorylation and the tricarboxylic acid cycle was observed in the LIP group (*Figure 1—figure supplement 2B, C*).

Gene Ontology (GO) Biological Process analysis of differentially expressed genes further demonstrated mitochondrial respiratory and electron transport dysfunction, as well as impaired oxidative phosphorylation in the gingiva of LIP mice, suggesting that mitochondrial dysfunction might contribute to cell senescence in periodontitis (*Figure 1—figure supplement 1D*). Meanwhile, upregulation of the cellular senescence pathway and a series of inflammatory-related pathways, including complement activation and response to lipopolysaccharide, were also enriched in the LIP group (*Figure 1—figure supplement 1D*). Besides that, the PI3K–AKT, MAPK, and NF-κB signaling pathways were also activated in the LIP group (*Figure 1—figure supplement 2D–F*), which were closely associated with the onset of cellular senescence and the secretion of SASP factors (*Raynard et al., 2022*; *Sayegh et al., 2024*; *Tang et al., 2023*). Collectively, these findings suggested that senescent cells gradually accumulated and senescence-related signaling pathways were activated during the progression of periodontitis.

## Gingival fibroblasts were the main cell type responsible for cellular senescence in periodontitis

To identify which cell types in periodontitis tissue are enriched for senescence, we re-analyzed public scRNA-seq data of healthy and periodontitis human gingiva (*Williams et al., 2021*). This data from 8 healthy and 13 periodontitis-affected gingival samples was analyzed, clustering the cells into 15 distinct groups (*Figure 2—figure supplement 1A*). These clusters were classified into fibroblasts, immune cells, epithelial cells, endothelial cells, and other cell types based on specific markers (*Figure 2A*, *Figure 2—figure supplement 1B*).

In periodontitis samples, there was a notable shift in cellular composition: immune cells increased while structural cells, such as fibroblasts, decreased (*Figure 2B, C*). Cellular senescence gene score analysis across different cell types revealed that fibroblasts in particular showed significant upregulation of senescence scores in periodontitis, indicating that they had the highest overall levels of senescence (*Figure 2D*). GSEA of differentially expressed genes between healthy and periodontitis fibroblasts further confirmed the activation of senescence pathways in periodontitis (*Figure 2E*). To

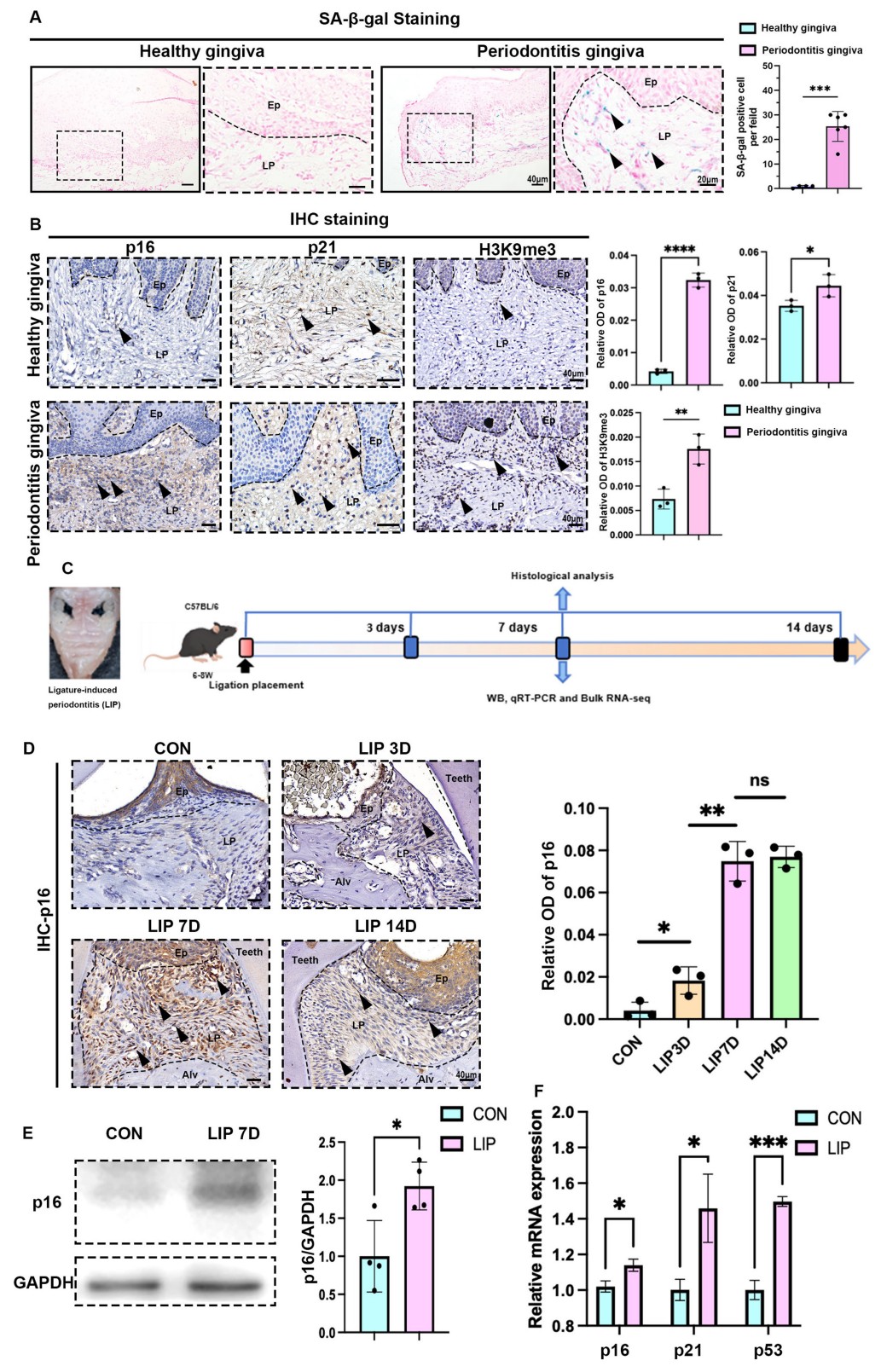

**Figure 1.** Characteristics of cellular senescence along with periodontitis progression. (**A**) Representative image of and semi-quantification of senescence-associated β-galactosidase (SA-β-gal) staining in healthy (*n* = 4 field) and periodontitis (*n* = 6 field) patient gingiva, scale bar = 40 or 20 µm. (**B**) Representative images of immunohistochemical (IHC) staining and semi-quantification of p16, p21, and H3K9me3 in healthy and

*Figure 1 continued on next page*

*Figure 1 continued*

periodontitis patient gingiva (*n* = 3 field), positive cells were indicated by black arrow, scale bar = 40 μm. (**C**) Analysis strategy of ligature-induced periodontitis (LIP) mouse model. (**D**) Representative image of IHC staining and semi-quantification of p16 in mouse gingiva of health and LIP post 3, 7, and 14 days (*n* = 3 field), scale bar = 40 μm. (**E**) Western blot images and semi-quantification of p16 protein levels in control (CON) and LIP post 7 days (LIP 7D) mouse gingiva (*n* = 4 independent experiments). (**F**) qrt-PCR analysis of p16, p21, and Tp53 in control (CON) and LIP 7D mouse gingiva (*n* = 3 independent experiments). Ep: epithelium; LP: lamina propria; Alv: alveolar bone; Teeth. Data are expressed as mean ± SD. *p < 0.05, **p < 0.01, ***p < 0.001, ****p < 0.0001.

The online version of this article includes the following source data and figure supplement(s) for figure 1:

**Source data 1.** Uncropped western blots with labeling for panel E.

**Source data 2.** Original tiff files of western blots for panel E.

**Figure supplement 1.** Bulk RNA-seq analysis of ligature-induced periodontitis (LIP) mice model.

**Figure supplement 2.** Bulk RNA-seq analysis of ligature-induced periodontitis (LIP) mice model in regard to cellular senescence.

---

further verify fibroblast senescence in periodontitis, we analyzed another dataset from GSE152042, which included samples from two healthy, one mild, and one severe periodontitis gingiva (*Caetano et al., 2021*). The results showed a decline in fibroblast proportion along with increasing disease severity (*Figure 2—figure supplement 1C, D*) and a corresponding increase in cellular senescence score (*Figure 2F*). Immunofluorescence (IF) staining on clinical sample confirmed that the proportion of p16[+] senescent fibroblasts in periodontitis rose to approximately 25%, compared to very few in healthy gingiva (*Figure 2G*). In vitro, healthy primary gingival fibroblasts (HGFs) stimulated with different concentrations of *Porphyromonas gingivalis* lipopolysaccharide (Pg-LPS) showed a dose-dependent increase in SA-β-gal-positive fibroblasts (*Figure 2—figure supplement 2A, B*). These findings suggest that gingival fibroblasts undergo significant senescence, potentially induced by Pg-LPS, during the progression of periodontitis.

## CD81[+] fibroblasts were identified as the major fibroblast subpopulation undergoing senescence

To examine the changes in gingival fibroblast subpopulations during periodontitis, we analyzed gingival fibroblasts from dataset GSE164241 (*Williams et al., 2021*) and identified seven distinct fibroblast subpopulations (*Figure 3A*). The cell proportion bar chart revealed an increase in subpopulations 0 in periodontitis compared to healthy controls (*Figure 3B*). We then applied a cellular senescence gene set (*Saul et al., 2022*) to score these subpopulations and found that subpopulation 0 exhibited the highest average expression levels, with a marked increase in periodontitis (*Figure 3C*). Gene Ontology (GO) enrichment analysis of the differentially expressed genes further confirmed that subpopulation 0 displayed upregulated aging characteristics (*Figure 3D*), indicating that this subpopulation is primarily responsible for fibroblast senescence. A density heatmap demonstrated that CD81 was predominantly enriched in subpopulation 0 (*Figure 3E*). The remaining subgroups were classified as EmFB (extracellular matrix-associated fibroblasts), P-EmB (pre-extracellular matrix-associated fibroblasts), MyFB (myofibroblasts), P-MyFB (pre-myofibroblasts), VFB (vascular-associated fibroblasts), and ImFB (immune-associated fibroblasts), based on GO analysis (*Figure 3F*). IF staining further showed that the proportion of CD81[+] fibroblasts in periodontitis increased to approximately 50%, compared to very few in healthy samples (*Figure 3G*). Thus, CD81[+] fibroblasts might represent a core senescent fibroblast population in human periodontitis.

## CD81[+] fibroblasts were terminally differentiating cells with high SASP expression

To investigate the role of fibroblasts in periodontitis-related inflammation, we analyzed the expression of SASP-related genes in each fibroblast group. CD81[+] fibroblasts exhibited elevated levels of SASP-related genes, including *IL-6*, *CXCL5*, *CXCL6*, *MMP1*, and *MMP3* (*Figure 4A*). Pseudotime analysis of fibroblast differentiation trajectories revealed that CD81[+] fibroblasts predominantly clustered at the end of the trajectory, indicating limited differentiation potential (*Figure 4B, C*). Functional enrichment analysis of genes showing gradual increases during differentiation highlighted pathways related to

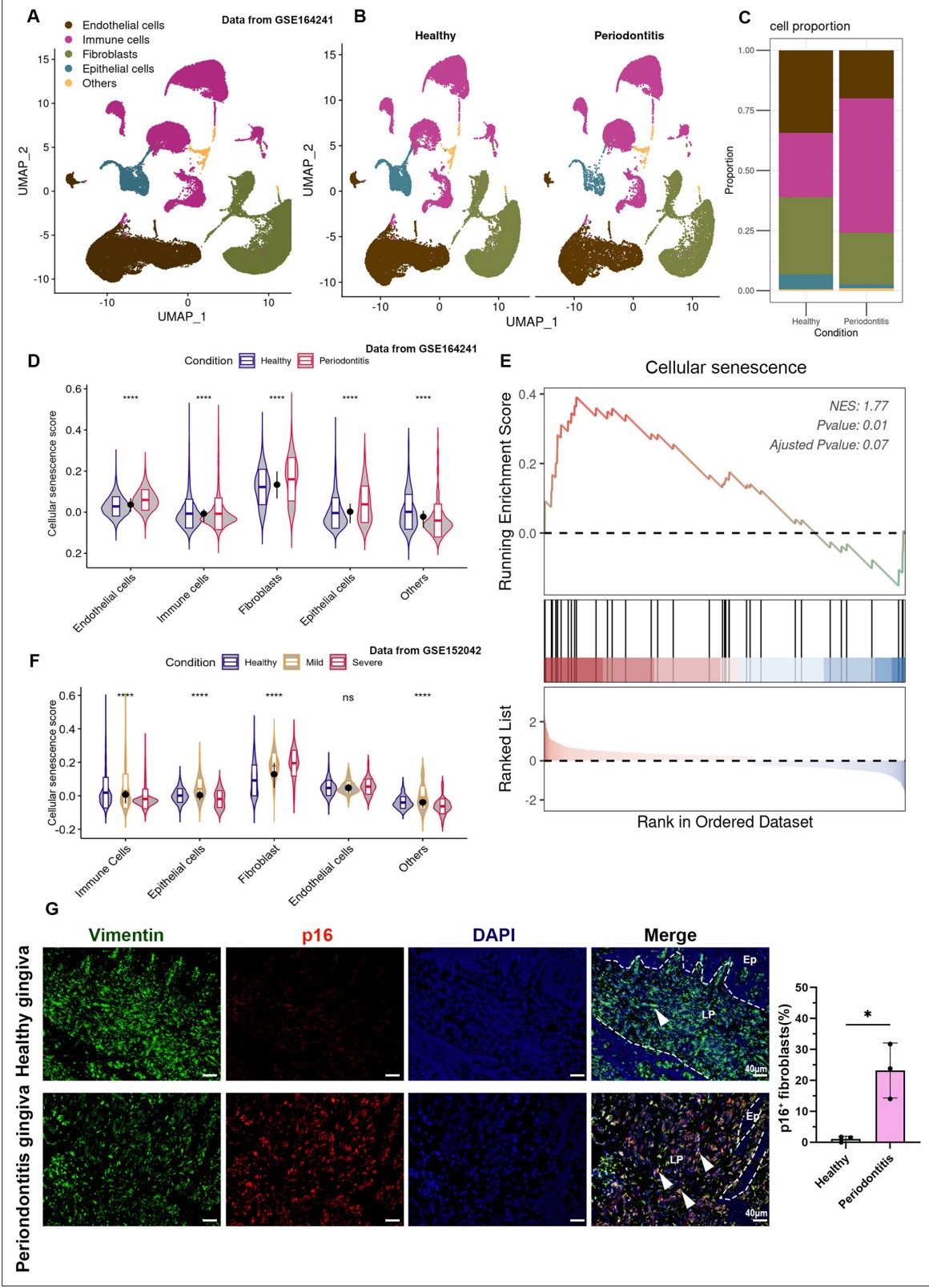

**Figure 2.** Cellular senescence of gingival fibroblasts in periodontitis. (**A**, **B**) UMAP diagram and single-cell annotation of cell clusters for the healthy and periodontitis patient gingiva from public dataset GSE164241. (**C**) Histogram of gingival tissue cell ratio in healthy and periodontitis patients. (**D**) The violin plot showing cellular senescence score of cell groups in healthy and periodontitis gingiva. (**E**) Gene set enrichment analysis (GSEA) of cellular senescence pathway in fibroblasts among periodontitis compared to those in healthy gingiva. (**F**) The violin plot showing cellular senescence score in

*Figure 2 continued on next page*

*Figure 2 continued*

subgroups in gingiva of healthy, mild, and severe periodontitis patients from public dataset GSE152042. (**G**) Immunofluorescence staining and semi-quantification of p16-positive fibroblasts in healthy and periodontitis patient gingiva. p16 (red), Vimentin (green), and nuclei (blue), Ep: epithelium; LP: lamina propria. White arrow indicates double positive cells, scale bar = 40 μm, *n* = 3.*p < 0.05, ****p < 0.0001.

The online version of this article includes the following figure supplement(s) for figure 2:

**Figure supplement 1.** Single cell RNA-seq analysis of healthy and peridontitis patient gingiva.

**Figure supplement 2.** SA-β-gal activity of human gingival fibroblasts stimulated by Pg-LPS.

inflammatory activation and aging characteristics (*Figure 4D*). Several SASP genes, including *CXCL1*, *CXCL6*, *CXCL8*, *IL6*, *SERPINE1*, *IGFBP4*, *MMP1* and *TIMP1*, also exhibited increased expression during differentiation (*Figure 4E*). Overall, our bioinformatics analysis demonstrated that CD81⁺ fibroblasts exhibited differentiation arrest and heightened expression of SASP factors, further implicating them in the inflammatory and senescent processes of periodontitis.

## CD81⁺ fibroblasts indirectly sustained inflammation by recruiting neutrophils via the C3/C3aR1 axis

To explore the communication between CD81⁺ fibroblasts and immune cells in periodontitis, we analyzed their interactions under diseased conditions. Our results revealed that CD81⁺ fibroblasts had the highest level of communication with immune cells, particularly neutrophils, compared to other fibroblast subgroups (*Figure 5A*). This suggests that CD81⁺ fibroblasts play a key role in mediating the immune response during periodontitis. Additionally, we observed a significant increase in the expression of C3 signaling pairs between CD81⁺ fibroblasts and neutrophil cells (*Figure 5B*). Previous studies have demonstrated the importance of sustained neutrophil infiltration in the progression of periodontitis, with C3 known to recruit neutrophils and contribute to the formation of neutrophil extracellular traps (*Ando et al., 2024*; *Kim et al., 2023*; *Song et al., 2023*). Further analysis of the C3 pathway showed that the C3 receptor–ligand pair was active in the communication between CD81⁺ fibroblasts and neutrophils in both healthy and periodontitis conditions (*Figure 5C*), underscoring its unique role in neutrophil recruitment. Notably, CD81⁺ fibroblasts exhibited the highest expression of the C3 ligand compared to other fibroblast subgroups, while the C3 receptor (C3aR1) was exclusively expressed by neutrophils in periodontitis (*Figure 5D*). We also detected higher *C3* expression in human periodontitis gingiva (*Figure 5E*), indicating its involvement in the disease. In vitro experiments further confirmed that periodontitis gingival fibroblasts secreted higher levels of C3 protein at 30 ng/ml compared to healthy fibroblasts at about 20 ng/ml (*Figure 5F*), and Pg-LPS stimulation could enhance C3 secretion by gingival fibroblasts from baseline at 10 to about 20 ng/ml (*Figure 5G*). Interestingly, spatial transcriptomic analysis of gingival tissue revealed that the regions expressing *CD81* and *SOD2*, a neutrophil marker, in periodontitis overlapped in the gingival lamina propria, showing a high spatial correlation (*Figure 5H*). These findings suggest that CD81⁺ fibroblasts might facilitate neutrophil infiltration through the C3/C3aR1 axis, contributing to the inflammatory response in periodontitis.

## Targeting cellular senescence in periodontitis could alleviate inflammation and bone resorption

In human periodontitis gingiva, we found that CD81⁺ fibroblasts might activate neutrophils via the C3/C3aR1 axis to exaggerate inflammation. To verify whether this mechanism exists in the LIP mouse model, we examined the expression of related markers. In the gingiva of the LIP model, p16⁺ fibroblasts, identified by p16 and Vimentin protein, comprised approximately 70% of total fibroblasts, significantly higher than the 10% observed in healthy mice (*Figure 5—figure supplement 1A*). Similarly, CD81⁺ fibroblasts accounted for about 30% of total fibroblasts, compared to less than 10% in the control group (*Figure 5—figure supplement 1B*). IF staining revealed co-localization of Vimentin, p16, and CD81 in LIP gingiva, indicating the presence of senescent CD81⁺ fibroblasts in the experimental periodontitis model (*Figure 5—figure supplement 1C*). We also observed a higher expression of C3 protein expression in the LIP group compared to controls (*Figure 5—figure supplement 2A, B*). Neutrophil infiltration, marked by MPO, increased from approximately 10% at baseline to 40% in the inflamed gingiva of LIP mice, notably in the epithelial and lamina layers (*Figure 5—figure supplement

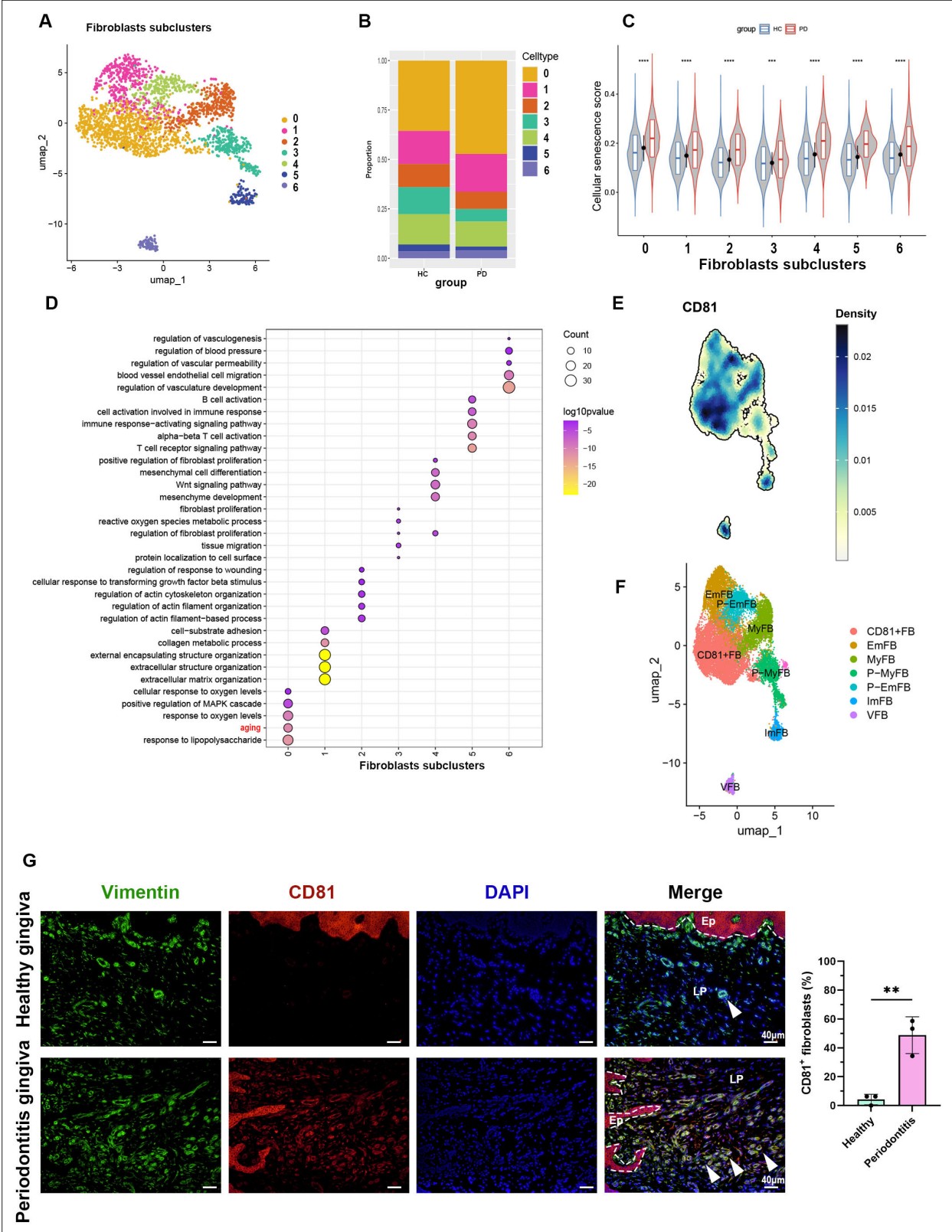

**Figure 3.** CD81 is identified as the potential marker of senescent gingival fibroblast. (**A**) UMAP diagram illustrated the cell subclusters of fibroblasts from public dataset GSE164241. (**B**) Histogram of fibroblasts subclusters ratio in healthy and periodontitis gingiva, respectively. (**C**) The violin plot showing cellular senescence score of each fibroblast subcluster in healthy and periodontitis gingiva. (**D**) Gene Ontology (GO) enrichment analysis of each fibroblast subcluster. Fibroblast subcluster 0 shows enrichment of aging process highlighted by red. (**E**) Density map of CD81 expression among

*Figure 3 continued on next page*

Figure 3 continued

fibroblast subclusters. (**F**) Re-annotation of fibroblast subcluster according to GO analysis. (**G**) Immunofluorescence staining and semi-quantification of CD81-positive fibroblasts in healthy and periodontitis patient gingiva. Vimentin (green), CD81 (red), and nuclei (blue), Ep: epithelium; LP: lamina propria. White arrow indicates double positive cells, scale bar = 40 μm, $n$ = 3. Data are expressed as mean ± SD. \*\*$p \leq 0.01$, \*\*\*\*$p \leq 0.0001$.

**2C, D**). Further staining demonstrated that $CD81^+$ $C3^+$ fibroblasts constituted the majority of fibroblasts in the LIP group (**Figure 5—figure supplement 2E, F**). Notably, $MPO^+$ neutrophils clustered around $CD81^+$ cells in the lamina of the LIP model (**Figure 5—figure supplement 2G**). These findings in the LIP mouse model suggest that $CD81^+$ fibroblasts with senescence characteristics might activate neutrophils through C3, similar to the mechanism observed in human periodontitis.

To further explore the role of senescent cells in periodontitis progression, we established an LIP mouse model treated with the senolytic drug ABT263, a Bcl2 inhibitor (**Figure 6A**). Hematoxylin and eosin (H&E) staining revealed that the ABT263-treated group exhibited reduced inflammatory cell infiltration in the gingiva compared to the vehicle control (**Figure 6B**). IHC staining of senescence markers p16 and H3K9me3 showed a significant reduction in senescent cells: $p16^+$ cells decreased from 20% to 8%, and $H3K9me3^+$ cells from 35% to 20%, after ABT263 administration (**Figure 6C, D, a, b**). To assess the effect of ABT263 on eliminating $CD81^+$ fibroblasts in periodontitis, IF staining demonstrated a drop in the proportion of $CD81^+$ fibroblasts from 40% to less than 20% after treatment (**Figure 6E, c**). Since our results suggested that $CD81^+$ fibroblasts might activate neutrophil infiltration via the C3/C3aR1 axis, we next evaluated the impact of senolytic treatment on C3 secretion and neutrophil infiltration. IHC analysis revealed a slight reduction in C3 intensity in gingival tissue and a significant decrease in the number of infiltrated neutrophils after ABT263 treatment (**Figure 6F, G, d, e**). Finally, we observed a marked reduction in the number of osteoclasts marked by CTSK in the ABT263-treated group, decreasing from 6 cells/mm² in the vehicle group to 1 cell/mm², which suggested less bone resorbing after ABT263 treatment (**Figure 6H, f**). Taken together, these results suggest that senolytic treatment with ABT263 could be a potential strategy to mitigate inflammation and bone resorption in periodontitis progression.

## Metformin alleviated the inflammation and bone resorption of periodontitis via inhibiting the interaction between $CD81^+$ fibroblasts and neutrophil cell

Metformin, an oral antihyperglycemic drug, has been preliminarily validated its therapeutic efficacy in periodontitis (**Neves et al., 2023**). However, the underlying mechanisms remain unclear. Increasing evidence suggests that metformin regulates cellular senescence, but its involvement in periodontitis-related senescence has yet to be reported (**Kodali et al., 2021**; **Soukas et al., 2019**). To uncover the role of metformin in periodontitis regarding cellular senescence, we first re-analyzed scRNA-seq data from GSE242714, which included the gingival tissue of periodontitis mice treated with metformin (**Neves et al., 2023**). Notably, we found that metformin treatment reduced the cellular senescence score of the periodontitis gingiva compared to untreated periodontitis gingiva (**Figure 7—figure supplement 1A**). Based on this, we further established an LIP mouse model and administered metformin daily for 14 days before and after the modeling time point to evaluate its effects on cellular senescence in periodontitis (**Figure 7A**). Micro-tomographic (micro-CT) imaging revealed that delayed bone loss around the periodontal area was found following metformin administration (**Figure 7B**), with the higher bone volume to tissue volume ratio (BV/TV, **Figure 7a**) and less distance of cement-to-enamel junction to alveolar bone crest (CEJ–ABC distance, **Figure 7b**) compared to LIP treated with $ddH_2O$ group. Histological analyses further demonstrated that metformin significantly mitigated periodontitis-induced inflammatory cell infiltration (**Figure 7—figure supplement 1B**), and collagen degradation (**Figure 7—figure supplement 1C, a**), as shown by H&E and Masson staining. Additionally, metformin reversed the upregulation of p16 (**Figure 7C, c**), p21 (**Figure 7—figure supplement 1D, b**), and H3K9me3 (**Figure 7—figure supplement 1E, c**) in the periodontitis model. Importantly, the number of $CD81^+$ fibroblasts was reduced in the LIP model after metformin administration compared to the untreated LIP group as well (**Figure 7D, d**). Furthermore, metformin reversed the elevated expression of C3 and MPO in periodontitis, compared to the periodontitis and $ddH_2O$ groups (**Figure 7E, F, e, f**). In vitro, senescent fibroblasts induced by Pg-LPS were treated with metformin (**Figure 7—figure supplement 2A**). Results showed that metformin decreased the

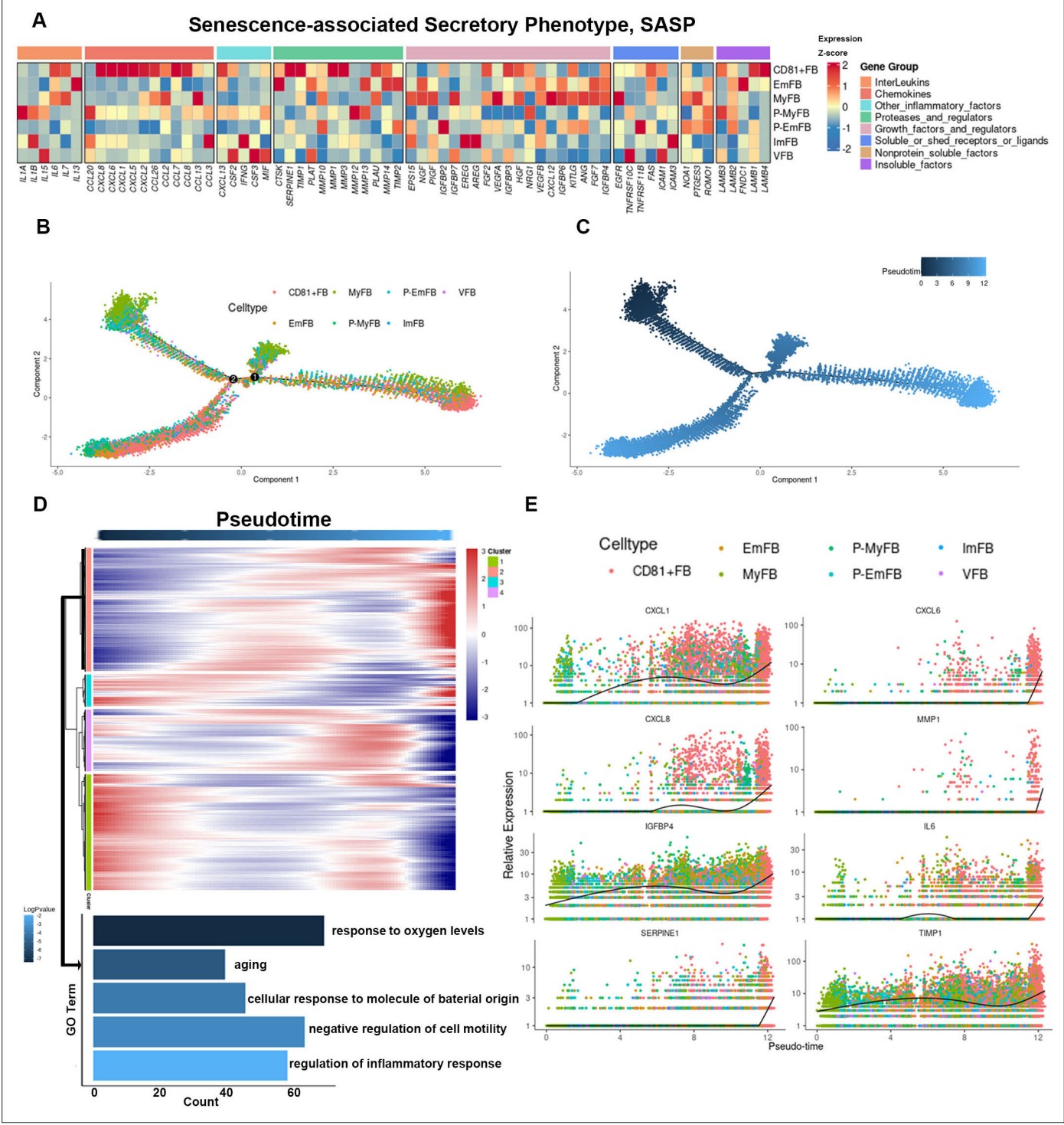

**Figure 4.** CD81[+] gingival fibroblasts are terminally differentiating cells with high senescence-associated secretory phenotype (SASP) genes expression. (**A**) Heatmap showing the relative expression for SASP genes in each fibroblast subcluster. (**B**) Trajectory reconstruction of each fibroblast subcluster. (**C**) Monocle pseudotime analysis revealing the progression of gingival fibroblast clusters. (**D**) Upper panel: Heatmap showing the scaled expression of differently expressed genes in trajectory as in (**C**), cataloged into four gene clusters (labels on left). Bottom panel: Gene Ontology (GO) analysis of expressed genes whose expression increases as the differentiation trajectory progresses. (**E**) SASP-related genes with increased expression as the differentiation trajectory progresses.

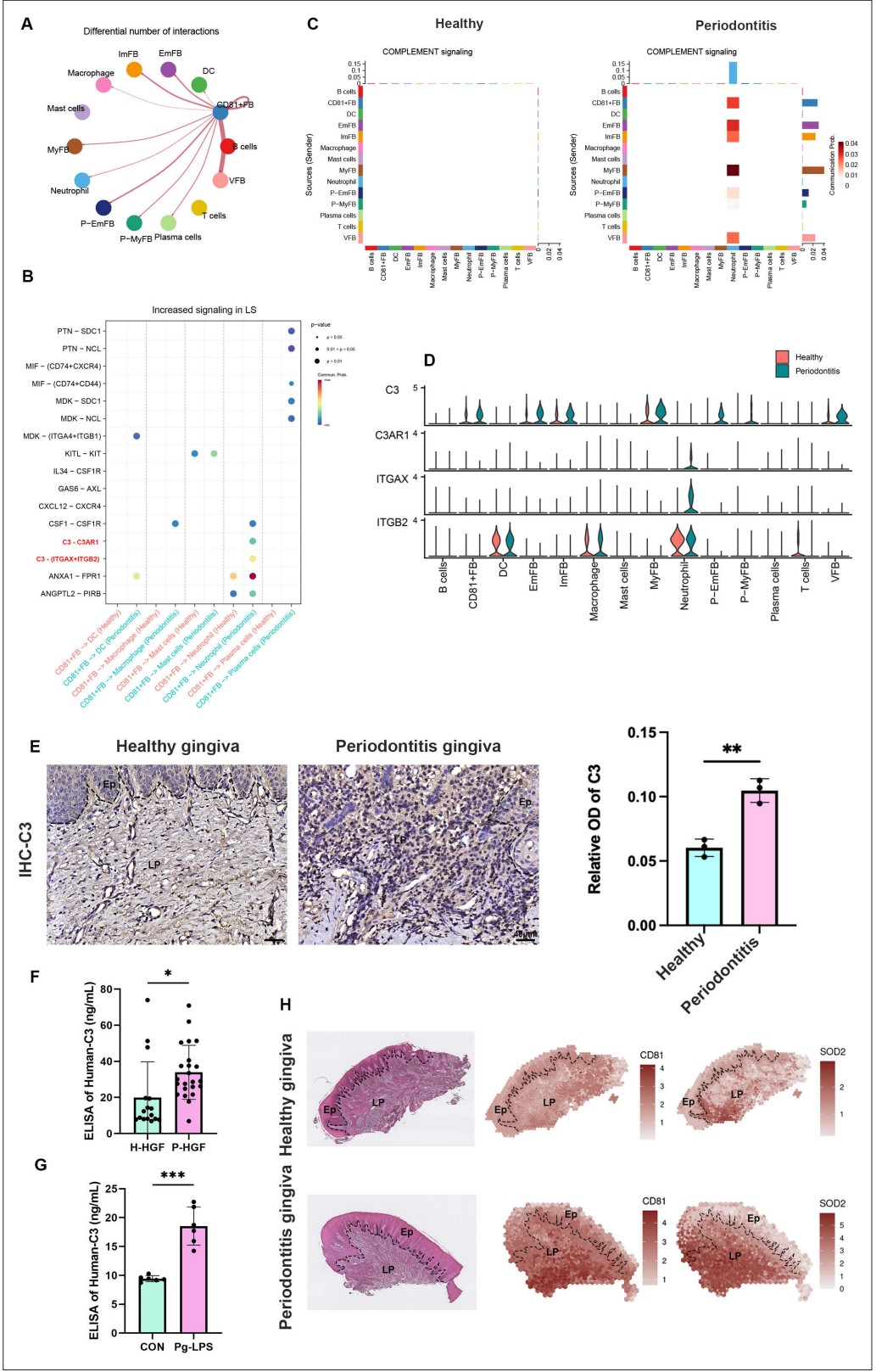

**Figure 5.** CD81[+] fibroblasts possibly recruit neutrophils via the C3/C3aR1 axis. (**A**) The relative number of interactions between CD81[+] fibroblasts and other cell types in periodontitis gingiva. (**B**) Significantly increased ligand–receptor interaction derived from CD81[+] fibroblasts. The C3–C3aR1 signaling axis increased between CD81[+] fibroblast and neutrophil, especially, which was highlighted by red. (**C**) The heatmap showing the

*Figure 5 continued on next page*

*Figure 5 continued*

communication patterns of the Complement signaling pathway between fibroblasts and immune cell type in healthy and periodontitis gingiva. (**D**) The expression level of four representative genes in Complement signaling pathway. (**E**) Representative image of and semi-quantification of immunohistochemical (IHC) staining regarding C3 in healthy and periodontitis gingiva. Scale bar = 40 μm, *n* = 3. (**F**) Enzyme-linked immunosorbent assay (ELISA) analysis of human-C3 secretion between healthy human gingival fibroblasts (H-HGF, *n* = 16 samples) and periodontitis human gingival fibroblasts (P-HGF, *n* = 23 samples). (**G**) ELISA analysis of human-C3 secretion in H-HGFs with (*Porphyromonas gingivalis* lipopolysaccharide [Pg-LPS] group) or without (CON group) 1 μg/ml Pg-LPS stimulated, *n* = 6 samples. (**H**) Hematoxylin and eosin (H&E) image and representative spatial mapping of CD81 and SOD2 in healthy and periodontitis gingiva from public dataset GSE206621. Co-localization in CD81 and SOD2, a neutrophil marker, was found in the periodontitis gingiva. Ep: epithelium; LP: lamina propria. Data are expressed as mean ± SD. *p ≤ 0.05, **p ≤ 0.01, ***p ≤ 0.001.

The online version of this article includes the following figure supplement(s) for figure 5:

**Figure supplement 1.** Identifcation of the CD81+p16+ fibroblasts in periodontitis mouse gingiva.

**Figure supplement 2.** C3 and MPO expression level in healthy or peridontitis mouse gingiva.

proportion of SA-β-gal-positive fibroblasts from 50% in the LPS group to 35% (*Figure 7—figure supplement 2B, C*). Metformin also reversed the protein expression of CD81, C3, and p16 in fibroblasts (*Figure 7—figure supplement 2D*). Additionally, metformin reduced the proportion of CD81/p16 and CD81/C3 double-positive gingival fibroblasts (*Figure 7—figure supplement 3A–D*). Collectively, these findings suggest that metformin alleviates inflammation and bone resorption in periodontitis by inhibiting the interaction between CD81+ fibroblasts and neutrophils, which provides a novel therapeutic strategy for periodontitis.

## Discussion

This study suggests that cellular senescence plays a role in the progression of periodontitis, and targeting cellular senescence may help alleviate the condition. We discovered that senescent gingival fibroblasts are associated with periodontitis pathology. Under continuous stimulation of Pg-LPS, oxidative stress caused by ROS accelerates cell senescence in gingival fibroblasts. These senescent cells highly express CD81, which contributes to the expansion of inflammation through pro-inflammatory metabolic activities and factors related to SASP. Additionally, they continuously recruit neutrophils through the C3 pathway, indirectly maintaining the inflammatory response. The use of Navitoclax and Metformin can slow down the progression of periodontitis by reducing gingival cell senescence (*Figure 8*).

The underlying mechanism of immune homeostasis instability and the transformation of chronic gingivitis into periodontitis has not been fully elucidated. Our findings will provide valuable insights for future studies on the pathological mechanism of periodontitis development. Gingival fibroblasts, which are essential cells in gingival connective tissue, have recently gained attention as key participants (*Wielento et al., 2023*). Previous studies have reported heterogeneity in gingival fibroblasts in periodontal tissues, with four subsets significantly altered in periodontitis: Fib 1.1 (CXCL1, CXCL2, CXCL13); Fib 1.2 (APCDD1, IGFBP2, MRPS6); Fib 1.3 (APOD, GSN, CFD); and Fib 1.4 (TIMP3, ASPN, COL11A1). Some of these clusters are directly associated with neutrophils and pro-inflammatory cytokines, suggesting that periodontal tissue immunity relies on strong matrix–neutrophil interactions within these tissues (*Williams et al., 2021*). Another study revealed the presence of genetic markers in a unique subgroup of gingival fibroblasts called AG fibroblasts (fibroblasts activated to guide chronic inflammation). These fibroblasts may have functional capabilities as oral immune surveillance agents and play a role in coordinating the initiation of gingival inflammation (*Kondo et al., 2023*). Caetano et al. conducted a study where they mapped stromal cells from healthy and periodontitis individuals. They identified a subset of fibroblasts that expressed ARGE pro-inflammatory genes at high levels (*Caetano et al., 2023*). In a more recent study, the team used multiomics techniques and fluorescence in situ hybridization to demonstrate the presence of a spatially restricted population of pathogenic fibroblasts in the gingival lamina propria. These fibroblasts expressed CXCL8 and CXCL10 and were responsible for recruiting neutrophils and lymphocytes in the periodontal pocket area. Additionally, they exhibited angiogenic properties (*Caetano et al., 2023*). The increasing

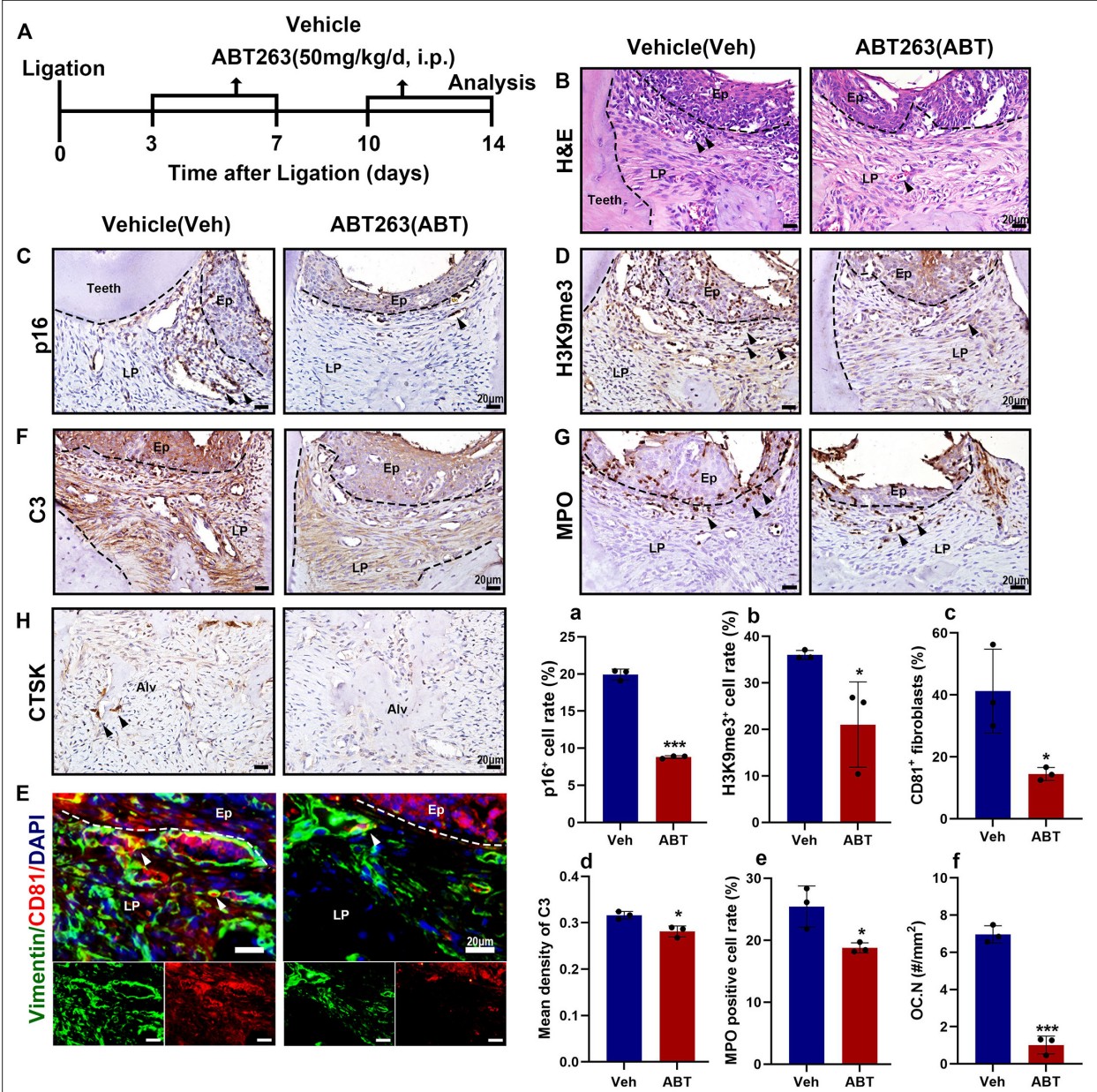

**Figure 6.** Senolytics therapy alleviates inflammation and bone resorption in the ligature-induced periodontitis (LIP) model. (**A**) Strategy of LIP mouse model treated by a senolytic drug Navitoclax. (**B**) Representative hematoxylin and eosin (H&E) staining image of each group, inflammatory cells were labeled by black arrows, scale bar = 20 μm. (**C**) Immunohistochemical (IHC) staining and (**a**) semi-quantification of p16 in each group, positive cells were labeled by black arrows, *n* = 3, scale bar = 20 μm. (**D**) IHC staining and (**b**) semi-quantification of H3K9me3 in each group, positive cells were labeled by black arrows, *n* = 3, scale bar = 20 μm. (**E**) Immunofluorescence staining and (**c**) semi-quantification of CD81 (red), Vimentin (green), and nuclei (blue) in control and LIP mouse gingiva, *n* = 3 mice, scale bar = 20 μm. White arrow indicates double positive cells. (**F**) IHC staining and (**d**) semi-quantification of C3 in each group, positive cells were labeled by black arrows, *n* = 3, scale bar = 20 μm. (**G**) IHC staining and (**e**) semi-quantification of MPO in each group, positive cells were labeled by black arrows, *n* = 3 field per group, scale bar = 20 μm. (**H**) IHC staining and (**f**) semi-quantification of CTSK in each group, positive cells were labeled by black arrows, *n* = 3 field per group, scale bar = 20 μm. Ep: epithelium; LP: lamina propria; Alv: alveolar bone; Teeth. Data are expressed as mean ± SD. *p ≤ 0.05, ***p ≤ 0.001.

amount of data supports the role of gingival fibroblast heterogeneity in the pathological mechanism of periodontitis, particularly in immune regulation (*Yin et al., 2025*). However, previous studies have mainly focused on immune disorders resulting from communication between fibroblasts and immune cells, neglecting the dynamic changes of fibroblasts themselves in periodontitis pathology. In this

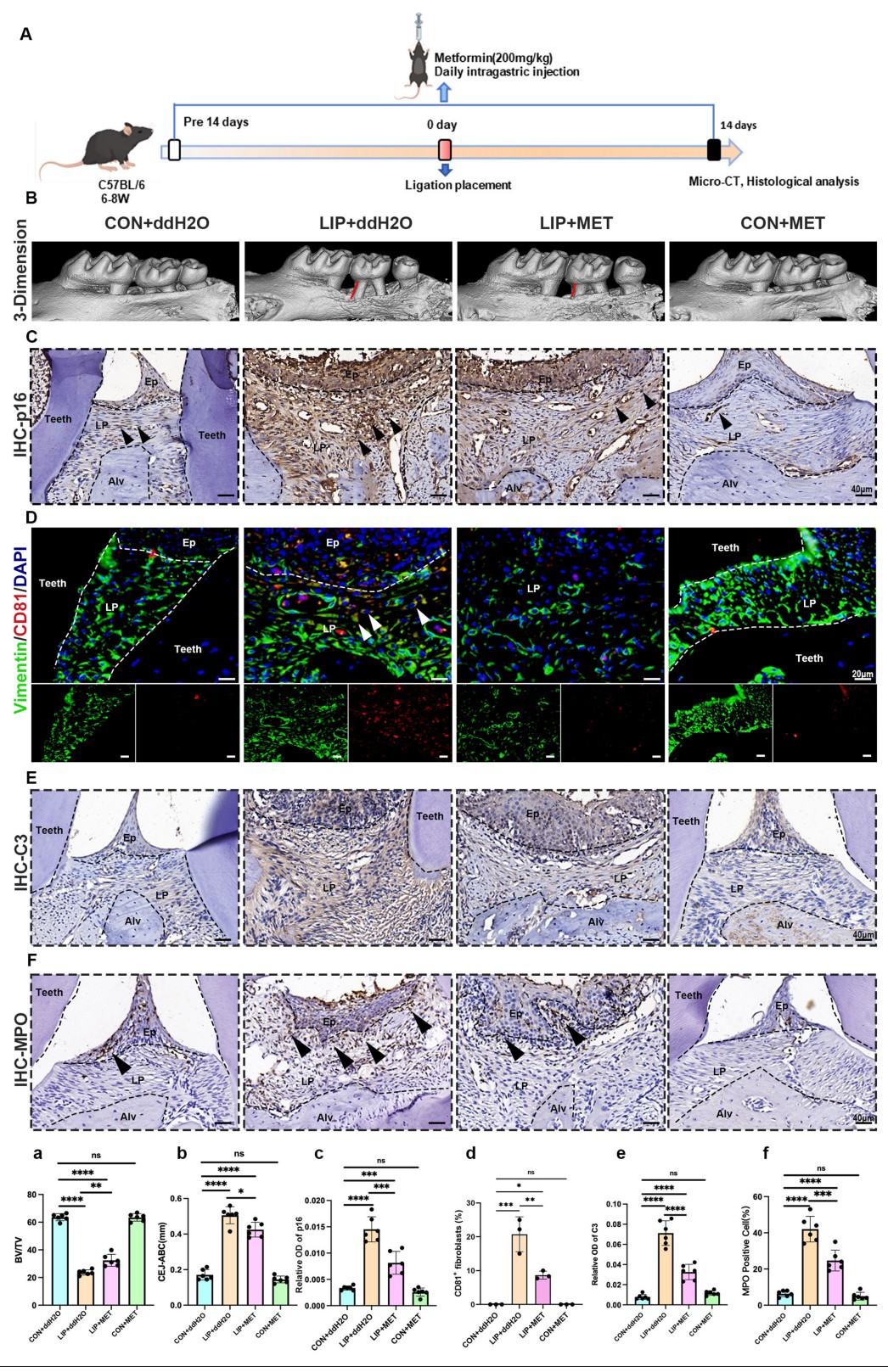

**Figure 7.** Metformin alleviates inflammation and bone resorption in the ligature-induced periodontitis (LIP) model via inhibiting the interaction between CD81+ fibroblasts and neutrophils. (**A**) Strategy of the LIP mouse model treated by metformin. (**B**) Three-dimensional (3D) visualization of the maxilla and quantified by the bone volume/tissue volume (BV/TV) ratio (**a**) the cement-to-enamel to alveolar bone crest (CEJ–ABC) distance

*Figure 7 continued on next page*

*Figure 7 continued*

(**b**) indicated by red line, *n* = 6 mice per group. (**C**) Immunohistochemical (IHC) staining and semi-quantification (**c**) of p16 in each group, positive cells were labeled by black arrows, *n* = 6 field per group, scale bar = 40 μm. (**D**) Immunofluorescence staining and (**d**) semi-quantification of CD81-positive fibroblasts in each group. Vimentin (green), CD81 (red), and nuclei (blue). White arrow indicates double positive cells, *n* = 3, scale bar = 20 μm. (**E**) IHC staining and (**e**) semi-quantification of C3 in each group, *n* = 6, scale bar = 40 μm. (**F**) IHC staining and (**f**) semi-quantification of MPO, a neutrophils marker, in each group, positive cells were labeled by black arrows, *n* = 6, scale bar = 40 μm. Ep: epithelium; LP: lamina propria; Alv: alveolar bone; Teeth. Data are expressed as mean ± SD. *p ≤ 0.05, **p ≤ 0.01, ***p ≤ 0.001, ****p ≤ 0.0001.

The online version of this article includes the following source data and figure supplement(s) for figure 7:

**Figure supplement 1.** Cellular senescence expression level of periodontitis gingiva after adminstration of the metformin.

**Figure supplement 2.** Cellular senescence expression level of Pg-LPS-stimulated HGFs after adminstration of the metformin.

**Figure supplement 2—source data 1.** Uncropped western blots with labeling for panel D.

**Figure supplement 2—source data 2.** Original tiff files of western blots for panel D.

**Figure supplement 3.** CD81+ p16+ and CD81+ C3+ fibroblasts in Pg-LPS-stimulated HGFs after adminstration of the metformin.

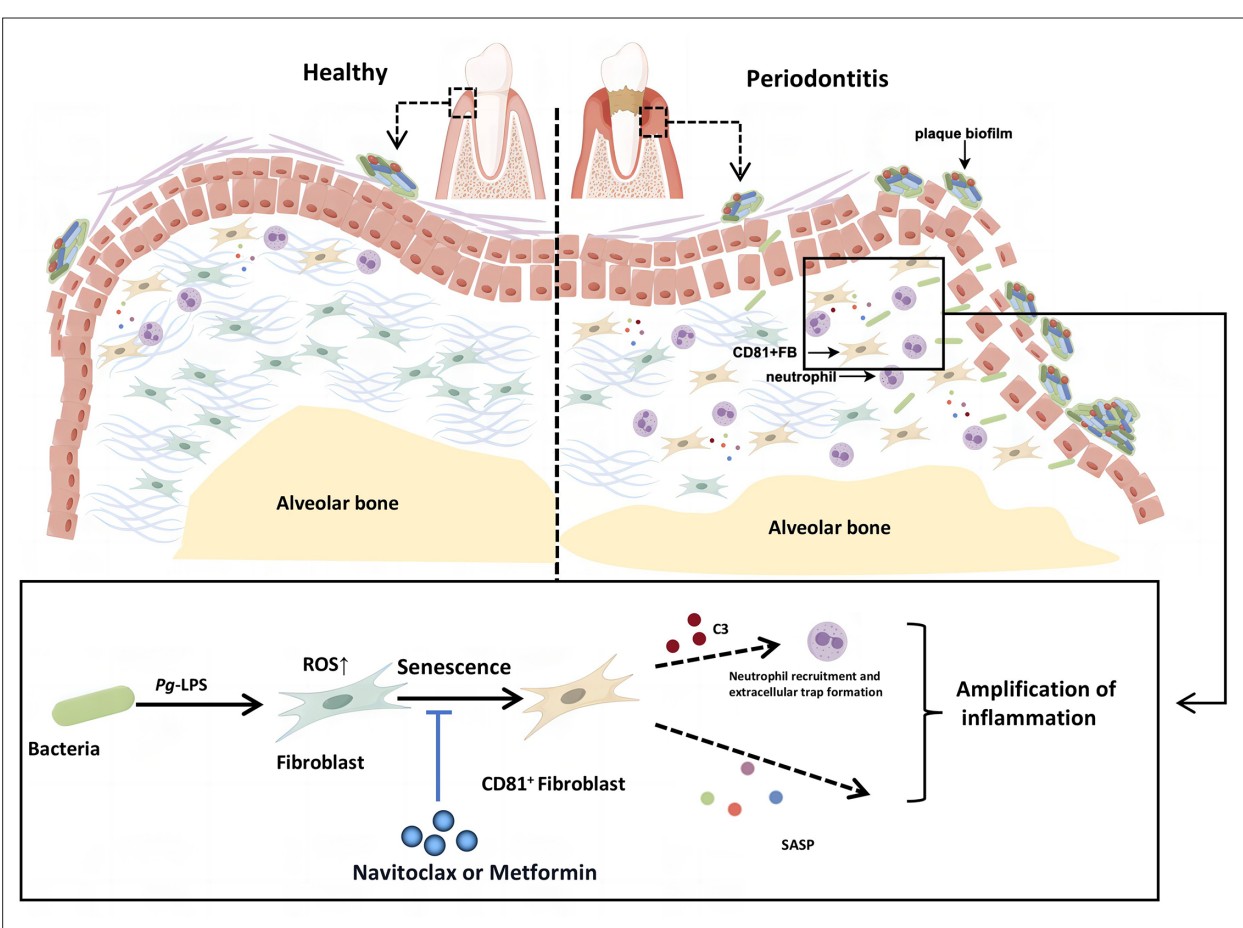

**Figure 8.** Schematic overview of the CD81+ senescent gingival fibroblast–neutrophil axis in periodontitis progression. We propose that the initial periodontal inflammation is triggered by the CD81+ senescent gingival fibroblast induced by bacterial virulence like *Porphyromonas gingivalis* lipopolysaccharide (Pg-LPS). CD81+ senescent gingival fibroblast could exaggerate inflammation in the periodontal tissue via secreting senescence-associated secretory phenotypes (SASPs) and recruiting neutrophils by C3. In addition, Navitoclax and Metformin could alleviate the cellular senescence of the fibroblast and rescue the uncontrolled inflammation and bone resorption.

study, we present a unique subset of fibroblasts with significantly altered gene signatures due to cell senescence, suggesting that cell senescence plays a crucial role in the heterogeneity of gingival fibroblasts.

It has been recognized that low concentrations of ROS produced during chronic inflammation can indirectly cause periodontal tissue destruction (*Chapple and Matthews, 2007*). Recent studies have also found that repeated exposure to LPS, a component of gram-negative bacterial membranes, leads to DNA damage in various cell types, including gingival and alveolar bone cells (*Aquino-Martinez et al., 2020*). Cells that survive from persistent DNA damage acquire a senescent phenotype, which in turn triggers the recruitment of immune cells through dysregulation of pro-inflammatory cytokines. Senescent cells often overexpress IL-6, IL-1α, IL-1β, and IL-8, collectively referred to as SASP (*Coppé et al., 2010*). Our findings indicate that gingival fibroblast senescence directly promotes the development of chronic periodontitis by secreting SASP-related factors, which may explain the formation of pro-inflammatory fibroblasts and their significant impact on immune regulation. Accumulating evidence suggests that drugs can regulate the activity of SASP, as demonstrated by *An et al., 2020*, who showed that short-term treatment with rapamycin can reduce gingival and alveolar bone inflammation and promote the regeneration of alveolar bone in elderly mice. Additionally, Kuang et al. reported that metformin inhibits the destructive effect of $H_2O_2$ on human periodontal ligament stem cells (PDLSCs), leading to a reduction in oxidative stress-induced aging (*Kuang et al., 2020*). Through oral administration of metformin, we have demonstrated its potential in alleviating the progression of periodontitis by delaying the senescence of gingival fibroblasts. However, further experiments are required to determine the decisive role of fibroblast senescence in periodontitis.

CD81, a member of the tetraspanin family of proteins, could serve as a cell surface marker (*Karam et al., 2020*) and a signaling pathway receptor (*Oguri et al., 2020*). CD81 is a major regulator of virus entry into cells and plays an important role in other pathogenic human viruses (*New et al., 2021*). Research on the role of CD81 has shown that it could form a complex with αV/β1 and αV/β5 integrins to activate the FAK signaling pathway (*Oguri et al., 2020*), which induces the interferon signaling pathway for immune response regulation (*Hanagata and Li, 2011*), and mediates NF-κB signaling pathway to induce IL-6 expression (*Ding et al., 2019*). Clinical studies have indicated a correlation between the level of CD81 in saliva and the severity of periodontitis disease (*Tobón-Arroyave et al., 2019*), as well as its association with the regulation of aging and inflammation (*Jin et al., 2018*). In our study, we observed that gingival fibroblasts with high CD81 expression exhibited a high enrichment of the NF-κB signaling pathway, leading to significant upregulation of IL-6 expression. The NF-κB pathway is recognized as a switch for cellular senescence, and NF-κB activation can drive cell senescence-related secretory phenotypes. Therefore, CD81 is likely to play a crucial role in regulating gingival fibroblast cell senescence. However, further investigation is needed to elucidate the specific molecular mechanism.

Finally, a link has been established between C3 from senescent fibroblasts and neutrophil infiltration in periodontitis. C3 has a strong recruitment ability for neutrophils and is crucial for the formation of neutrophil extracellular traps (NETs) (*Yipp et al., 2012*). Persistent neutrophil infiltration and hyperresponsiveness, including the formation of NETs, play significant roles in the development of periodontitis (*Uriarte and Hajishengallis, 2023*). Genetic analysis and preclinical studies have confirmed C3 as a potential pharmacological target for periodontitis treatment (*Alayash et al., 2024*; *Hajishengallis and Chavakis, 2021*). Gingival fibroblasts stimulated with IFN-γ upregulated the expression of chemokines (CXCL9, -10, -11, CCL8), molecules involved in antigen presentation, complement component 3 (C3), and other immune response-related molecules (*Ha et al., 2022*). Our experimental results have demonstrated that CD81[+] gingival fibroblasts are an important source of C3. Understanding the source and mechanism of C3 complement in periodontitis is of great significance for comprehending the pathological development of the disease and can provide a new perspective for designing drug schemes.

Our study focused on identifying a specific group of gingival fibroblasts that express high levels of CD81 during the development of periodontitis. Our findings suggest that these CD81[+] gingival fibroblasts exhibit characteristics of cellular senescence and possess strong pro-inflammatory abilities. Furthermore, we have established a connection between CD81[+] gingival fibroblasts and the recruitment and hyperactivation of neutrophils through C3. However, further investigations are required to explore the association between CD81 and cellular senescence, as well as its potential as

a therapeutic target. In conclusion, our research provides valuable insights and treatment strategies for understanding the progression of periodontitis.

## Materials and methods

### Human samples

All individuals provided written informed consent, and this study was approved by the Ethics Committee of School & Hospital of Stomatology Wuhan University (WDKQ2024B01). A total of 16 participants were recruited in this study (healthy group: *n* = 8; periodontitis group: *n* = 8). The basic information of the included patient is listed in *Supplementary file 1A*. Healthy control group included patients who underwent wisdom tooth extraction or crown lengthening procedures, and inclusion criteria are as follows: (1) age 18–65 years old; (2) good general health, no systemic diseases, able to tolerate periodontal surgery; (3) no erythema, edema, bleeding, and other symptoms in gingival tissue; (4) no use of nicotine-related products in the recent 6 months. The periodontitis group included patients who went through pocket reduction surgeries. Inclusion criteria for patients with chronic peri-odontitis were as follows (*Armitage, 1999*): (1) age 18–65 years; (2) good general health, no systemic disease, and tolerance to periodontal surgery; (3) mild gingival tissue redness, bleeding on probing, or clinical attachment loss ≥ 4 mm or probing depth ≥ 5 mm in non-acute inflammatory periods; (4) no use of nicotine-related products in the last 6 months. Collected gingiva were used for primary cell culture and histological analysis in this study.

### Primary gingival fibroblast cell culture isolation and culture

Collected gingiva tissues were transported from the clinic to the laboratory in pre-cooled phosphate-buffered saline (PBS) solution and rinsed with PBS several times to remove debris. And then, the tissues were minced into small fragments with a diameter of approximately 1–3 mm. The tissue pieces were digested with 2 µg/ml type II collagenase (2275GR001, BioFroxx, Germany) at 37°C for 2 hr, and collected cell precipitates were incubated for 5–7 days at 37°C and 5% $CO_2$ in DMEM high-glucose medium (DMEM, YC-2067, China) supplemented with 20% fetal bovine serum (PAN-SERATECH, South America) (*Li et al., 2024*). The primary gingival fibroblast cells that grew out of the explants were cultured and passaged. Primary gingival fibroblasts at passages four to eight were used in the following experiments. Gingival fibroblasts derived from healthy gingiva were labeled as H-HGF, while those derived from periodontitis gingiva were labeled as P-HGF.

### Pg-LPS-induced HGFs treated with metformin

To investigate the effect of Pg-LPS on the cellular senescence of gingival fibroblasts, healthy HGFs were seeded at 5000 per well in 96-well plate and incubated in complete medium at 37°C overnight. And then, the HGFs were stimulated by Pg-LPS (InvivoGen, USA) at 0, 0.5, 1, 5, and 10 µg/ml for 24 hr. At last, the samples were used for SA-β-gal staining.

To evaluate the effect of metformin on the cellular senescence of gingival fibroblasts stimulated by Pg-LPS, HGFs at 150,000 cells per ml using hemacytometer were seeded in 3 ml plates and incubated in complete medium at 37°C overnight. For the LPS + MET group, cells were pre-treated with metformin (HY-B0627, MedChemExpress, China) at 2 mM for 24 hr. And then, for the LPS and LPS + MET group, cells were stimulated with Pg-LPS (InvivoGen, USA) at 1 µg/ml for another 24 hr according to a previous study (*Sun et al., 2023*). Subsequently, HGF cells were harvested for subsequent SA-β-gal staining, western blot analysis, and IF staining.

### Enzyme-linked immunosorbent assay analysis of C3

HGFs were seeded in 6-well plates with 2 ml complete cell culture. When it comes to 80 or 90 % cell confluency, cells were kept in a resting state for 24 hr in serum-free medium. The supernatant of cell culture was collected after centrifugation at 12,000 rpm for 20 min. The concentration of C3 in cell culture supernatants was assessed by Human C3 ELISA kit (ELK1059, ELK Biotechnology, China) according to the manufacturer's instruction.

### Staining for SA-β-gal

SA-β-gal staining was performed using the Senescent β-Galactosidase Staining Kit (C0602; Beyotime Biotechnology, China) according to the manufacturer's instructions. Cell samples were incubated for

12 hr while tissue samples were incubated for 24 hr at 37°C in a $CO_2$-free temperature chamber. Tissue sections were then stained by nuclear red staining solution. Positive cells were blue-stained and all cells were nuclear red-stained. Three randomized regions of interest were captured under an ordinary light microscope (DP72 microscope, Olympus, Japan) and the percentage of positive cells was counted by ImageJ v2.0 (NIH, Bethesda, MD, USA).

## LIP mouse model treated by senolytics or metformin

C57BL/6 mice (8 weeks, male) were purchased from Hubei Provincial Center for Disease Control and Prevention and bred in specific pathogen-free animal laboratory of the School & Hospital of Stomatology, Wuhan University. The animal experiments were conducted according to the ARRIVE guidelines 2.0. Animals were approved by the Animal Research Ethics Committee at the School & Hospital of Stomatology, Wuhan University, China (No. S07922040A). The animals were housed in an SPF environment with controlled temperature/humidity with 12 hr light/dark cycle.

To investigate the role of senescent cells in periodontitis progression, the LIP mouse model was treated by senolytics drug ABT263 (HY-10087, MedChemExpress, China). In brief, after anesthetics, the mice were ligated with a 5-0 silk (SA82G, ETHICON, China) between the maxillary first and second molars and knots were tied on the palatal side to secure the ligature. The ligatures were examined daily to ensure that they remained in place during the experimental period. LIP mice were divided into two groups: Vehicle and ABT263 group. Each group included six mice. LIP mice were intraperitoneally injected with vehicle alone (10% DMSO + 40% PEG300 + 5% Tween-80 + 45% Saline) or with ABT263 (50 mg/kg/day; HY-10087; MedChemExpress, China) as previously (*Li et al., 2023*). Three days after ligation, vehicle and ABT263 were given to mice for two cycles of 4 consecutive days, with 3 days of rest between cycles. After 14 days post-ligation, mice were euthanized, and their maxilla and gingiva were collected for histological staining.

To investigate the effect of metformin on the periodontitis progression, the LIP mouse model was treated by metformin (HY-B0627, MedChemExpress, China). Mice were allocated into four groups: CON + ddH2O group, LIP + ddH2O group, LIP + MET group, and CON + MET group, each group included six mice. LIP + MET group and CON + MET group were treated with 200 mg/kg metformin while CON + ddH2O group and LIP + ddH2O were treated with the distilled water as the control. Metformin or ddH2O was given by intragastric administration once a day for 14 days before LIP model establishment. On the 15th day after intragastric administration, LIP + ddH2O group and LIP + MET group were ligated with a 5-0 silk between the maxillary left first and second molars and knots were tied on palatal side to secure the ligature. A second set of controls included mice that were not treated with ligatures on either side. Metformin or ddH2O was given once a day for another 14 days. At the end of the time frame, mice were euthanized and their maxilla and gums were collected for micro-CT and histological analysis.

## Micro-CT scanning and analysis

Micro-CT scanning was performed using Bruker Micro-CT SkyScan1276 (Konitich, Germany). The region of interest (ROI) was established in a three-dimensional (3D) scope: vertically, starting from 0.2 mm apical to the CEJ of the second molar (2nd M), extending toward the root apical to get a span of 0.5 mm; mesiodistally, ranging from the most mesial aspect of the CEJ of the first molar (1st M) to the root furcation of the third molar (3rd M); buccolingually and lingually, ranging around the root furcation of the 2nd M within a span of 1.5 mm. The ratio of BV/TV was calculated based on this ROI. The distances between the CEJ and the ABC were measured at the 2nd M. The 3D reconstruction, calculation, and measurements were conducted using the CTAn software (version 1.18.8.0, SkyScan, Germany). All measurements were repeated three times with 6 mice per group, with the average value of the bilateral maxillary alveolar bone taken as one sample for statistical analysis.

## Protein extraction and western blot

Protein extracted from mice samples or primary gingival fibroblasts was dissolved in 80 µl of RIPA buffer to extract total protein, supplemented with protease and 1% phosphatase inhibitors. All samples were quantified and normalized using a protein assay kit known as bicinchoninic acid (Thermo Fisher Scientific, Waltham, MA, United States). Following a 10-min heat treatment at 95°C, the samples underwent sodium dodecyl sulfate–polyacrylamide gel electrophoresis for separation and were then transferred

to a polyvinylidene fluoride membrane (Millipore). The membrane was blocked using the primary antibody-blocking solution and then incubated overnight at 4°C with primary antibodies against p16 (10883-1-AP, Proteintech, China), CD81 (66866-1-IG, Proteintech, China), β-actin (66009-1-Ig, Proteintech, China), C3 (21337-1-AP, Proteintech, China), and GAPDH (PMK052S, Biopm, China). Subsequently, the membrane was treated with horseradish peroxidase-conjugated secondary antibodies at 37°C for 1 hr. Visualization of signals was conducted using an Ultrasensitive ECL Detection Kit (Thermo Fisher Scientific, Waltham, MA, United States) with the ChemiDoc MP Imaging Systems (Bio-Rad, USA). Protein levels were normalized to β-actin or GAPDH using ImageJ analysis software.

## RNA extraction and RT-qPCR

To extract total RNA, the Trizol reagent and standard collection procedure were utilized. Total RNA concentration was measured using a Nanodrop2000 instrument (Thermo Fisher Scientific, Waltham, MA, United States). According to the guidelines provided by the manufacturer, the total RNA was subjected to reverse transcription into cDNA using the HiScript II Q RT SuperMix (Vazyme). The amplification reaction was performed using ChamQ SYBR qPCR Master Mix (Vazyme) in the QuantStudio 6 Flex System (Thermo Fisher Scientific, Waltham, MA, United States). The primers for the experiment were bought from Sangon Biotech Co., Ltd. The results were analyzed using the $2^{-\Delta\Delta Ct}$ method, with normalization to β-actin and calibration to the control group. The forward and reverse primer sequences of the target genes used in the experiment can be found in *Supplementary file 1B*.

## Histological analysis

The human gingiva samples were kept in 4% paraformaldehyde for 24 hr, followed by dehydration and fixation in paraffin or optimal cutting temperature compound. The mice maxilla with gingival tissues were kept in 4% paraformaldehyde for 24 hr, followed by 4 weeks of decalcification with 15% EDTA at pH 7.4. The decalcifying solution underwent replacement every 2 days. Tissues were then dehydrated, fixed in paraffin, and sectioned. The sections were stained by H&E, Masson, IHC, and IF staining. IHC and IF staining were performed according to the manufacturer's instructions (MXB Biotechnologies, Fuzhou, China). The primary antibodies used for immunohistochemistry included p16 (1:1000; Cat: 10883-1-Ap, Proteintech, China), p21 (1:200, Cat: 10355-1-AP, Proteintech, China), H3K9me3 (1:1000. Cat: M1112-3, HUABio, China), C3 (1:200, Cat: 21337-1-AP, Proteintech, China), MPO (1:200, Cat: Ab208670, Abcam), and CTSK (1:200, Cat: 121071, Proteintech, China). IF staining was performed with the antibodies of CD81 (1:1000, Cat: 10883-1-AP, Proteintech, China), Vimentin (1:200, Cat: A19607, ABclonal, China), p16, C3, and MPO as previously described. In IHC staining, 3,3-diaminobenzidine tetrahydrochloride (Zhongshan Biotechnology, Ltd, China) was utilized for visualization. For double IF staining, anti-mouse and rabbit secondary antibodies had Cy3 red and 488 nm green fluorescent markers (ABclonal, China). For triple IF staining, the nucleus of cells in tissues was stained using DAPI (Zhongshan Biotechnology, Ltd, China). The stained sections were examined and captured using an Olympus DP72 microscope (Olympus Corporation, Japan). For semi-quantification of protein expressions, the mean optical density of positive stains was measured using the imageJ2 software (version: 2.14.0, National Institutes of Health, Bethesda, MD). For the semi-quantification of Masson's trichrome, the collagen volume fractions (stained blue) for individual sections were measured using ImageJ2 software.

## Bulk RNA sequencing

For bulk RNA sequencing, RNA was extracted using the methods outlined in the qRT-PCR protocol. The total RNA was then sent to the Analysis and Testing Center at the Institute of Hydrobiology, Chinese Academy of Sciences (Wuhan, China) for quality control, library preparation, and sequencing on the Illumina platform. We utilized the Illumina TruSeq RNA library preparation kit, which generated libraries with insert fragment lengths of approximately 400–500 bp. The resulting fastq reads were aligned to the mouse genome (GRCm38) using a dedicated RNA-seq aligner. We filtered the raw data quality using Trimmomatic (version 0.36) (*Bolger et al., 2014*). The filtered reads were subsequently aligned to the reference genome with HISAT2 (version 2.2.1), and the aligned reads were quantified using StringTie (*Pertea et al., 2016*). The average mapping rate for each sample exceeded 90%, with sequencing depths ranging from 30 to 40 M reads.

We employed DESeq2 (version 1.34.0) to identify differentially expressed gene sets, applying thresholds of $|\log_2(\text{fold change})| > 1$ and a significance level of $p < 0.05$ (*Love et al., 2014*). The selected differentially expressed genes were then subjected to GO enrichment analysis. Additionally, GSEA was performed using GSEA_Linux_4.1.0 to identify relevant pathways (*Subramanian et al., 2005*). Significant gene sets were determined based on three criteria: $p < 0.05$, false discovery rate (FDR)<0.25, and an absolute normalized enrichment score >1.

## Single-cell RNA sequencing analysis

Single-cell RNA transcriptome including GSE164241, GSE152042, and GSE242714 was obtained from the GEO dataset. GSE164241 contained 70,407 cells from 13 healthy samples and 8 periodontitis samples (*Williams et al., 2021*). GSE152042 contained 12,379 cells from 2 healthy samples, 1 periodontitis sample (mild), and 1 periodontitis sample (severe) (*Caetano et al., 2021*). GSE242714 contained 6473 cells from the control mice and LIP mice samples, which were put on either water or Metformin samples ($n = 5$ group) (*Neves et al., 2023*). As for scRNA-seq, the 'Seurat4.4.0' package was applied to integrate different samples with CCA (cross-dataset normalization) method. GSE164241 cell profiles were filtered with criteria of Feature_RNA >200 & nFeature_RNA <5000 & MT_percent <10 & nCount_RNA <25,000 & nCount_RNA >1000, then GSE152042 cell profiles were filtered with criteria of Feature_RNA >500 & nFeature_RNA <6000 & MT_percent <20, and GSE242714 cell profiles were filtered with criteria of Feature_RNA >300 & nFeature_RNA <5000 & MT_percent <15 & nCount_RNA <25,000 & nCount_RNA >500, then those data were further normalized using the 'LogNormalize' method, and the unique gene markers in each group were identified with the 'FindMarkers' function. 'UMAP' was used to display the cell distribution. The function 'AddModuleScore' was used to reflect differences in biological processes in different cell populations.

## Fibroblast cell re-clustering analysis

Fibroblast clusters from GSE164241 were re-analysed and were then re-normalized by calling the 'NormalizeData' function to account for the reduction in cell numbers subsequent to subsetting the data. The top 2000 most variable features across the dataset were then identified using the 'FindVariableFeatures'. These variable features were subsequently used to inform clustering by passing them into the 'RunPCA' command. Via 'Elbowplot', we identified that the first eight principal components should be used for downstream clustering when invoking the 'FindNeighbors' and 'RunUMAP', as detailed above. Harmony was applied to correct for inter-sample variation, and the Harmony components were used for clustering and UMAP embedding.

## Gene function enrichment analysis

GO analysis was performed using 'Enrichr' on the top 200 differentially expressed genes (adjusted p-value <0.05 by Wilcoxon rank sum test) (*Kuleshov et al., 2016*). GO terms shown are enriched at FDR <0.05. The enrichment analysis between different fibroblast subsets in scRNA-seq was performed by 'Metascape' and further drawn by the 'ggplot2' package. Four methods, including 'ssGSEA', 'AUCell', 'UCell', and 'singscore' were used for enrichment analysis between different clusters. Images were further drawn by 'irGSEA'. GSEA was applied to validate the result on RNA-seq with default settings (1000 permutations for gene sets, Signal2Noise metric for ranking genes).

## 'Cellular senescence' and 'senescence-associated secretory phenotypes' gene set

The cellular senescence gene set, which was used to reflect the degree of cellular senescence, has been validated across species in a variety of cell lines and multiple sequencing data including scRNA-seq, bulk RNA-seq, etc. It has better validation efficiency than previously known gene sets associated with cell senescence (*Saul et al., 2022*). SASP includes several soluble and insoluble factor families. These factors can affect surrounding cells by activating various cell surface receptors and corresponding signal transduction pathways that may lead to a variety of pathologies. SASP factors can be divided globally into the following main categories: soluble signal transduction factors (ILs, chemokines, and growth factors), secreted proteases, and secreted insoluble protein/extracellular matrix components (*Coppé et al., 2010*).

## Pseudotime trajectory analysis

We applied the single-cell trajectory analysis utilizing Monocle2 using the DDR-Tree and default parameter. Before Monocle analysis, we selected marker genes from the Seurat clustering result and raw expression counts of the cell passed filtering. Based on the pseudotime analysis, branch expression analysis modeling (BEAM Analysis) was applied for branch fate determined gene analysis (*Qiu et al., 2017*).

## Cell–cell communication analysis

The cell–cell communication was measured by quantification of ligand–receptor pairs among different cell types. Gene expression matrices and metadata with major cell annotations were used as input for the CellChat package (v1.6.1) (*Jin et al., 2021*).

## Spatial transcriptomics data analysis

Spatial transcriptomics slides were printed with two identical capture areas from one healthy sample and one periodontitis sample (*Caetano et al., 2023*). The capture of gene expression information for ST slides was performed by the Visium Spatial platform of 10x Genomics through the use of spatially barcoded mRNA-binding oligonucleotides in the default protocol. Raw UMI counts spot matrices, imaging data, spot-image coordinates, and scale factors were imported into R using the Seurat package (versions 4.2.2). Normalization across spots was performed with the 'LogVMR' function. Dimensionality reduction and clustering were performed with independent component analysis (PCA) at resolution 1 with the first 30 PCs. Signature scoring derived from scRNA-seq or ST signatures was performed with the 'AddModuleScore' function with default parameters in Seurat. Spatial feature expression plots were generated with the SpatialFeaturePlot function in Seurat (versions 3.2.1). To further increase data resolution at a subspot level, we applied the BayesSpace package (*Zhao et al., 2021*).

## Statistical analysis

GraphPad Prism software (version 6.0, USA) was used for statistical analyses. Data were presented as the mean and SD in all graphs. Data were analyzed using the unpaired Student's *t*-test in order to compare group pairs or ANOVA for multiple group comparisons. Statistical significance was set at $p < 0.05$.

# Acknowledgements

Thanks to all clinical participants for their contribution. We would like to thank Dr. Zhixian Qiao and Xiaocui Chai at The Analysis and Testing Center of Institute of Hydrobiology, Chinese Academy of Sciences for their assistance with RNA-seq and data analysis. The National Natural Science Foundation of China (No. 32370816) and Research Project of School and Hospital of Stomatology Wuhan University (No. ZW202403) for Haibin Xia. Undergraduate Training Programs for Innovation and Entrepreneurship of Wuhan University (No. S202510486507) for Min Wang. The funders had no role in study design, data collection, and interpretation, or the decision to submit the work for publication.

# Additional information

### Funding

| Funder | Grant reference number | Author |
|---|---|---|
| National Natural Science Foundation of China | No. 32370816 | Haibin Xia |
| Research Project of School an Hospital of Stomatology Wuhan University | No. ZW202403 | Haibin Xia |
| Undergraduate Training Programs for Innovation and Entrepreneurship of Wuhan University | No. S202510486507 | Min Wang |

| Funder | Grant reference number | Author |
|--------|------------------------|--------|

The funders had no role in study design, data collection, and interpretation, or the decision to submit the work for publication.

## Author contributions

Liangliang Fu, Conceptualization, Resources, Data curation, Software, Formal analysis, Validation, Investigation, Visualization, Methodology, Writing – original draft; Chenghu Yin, Data curation, Formal analysis, Validation, Investigation, Visualization, Methodology, Writing – original draft; Qin Zhao, Ting Xia, Quan Sun, Resources, Data curation, Methodology, Writing – review and editing; Shuling Guo, Wenjun Shao, Software, Visualization, Methodology, Writing – review and editing; Liangwen Chen, Formal analysis, Validation, Writing – review and editing; Jinghan Li, Resources, Software, Writing – review and editing; Min Wang, Resources, Supervision, Funding acquisition, Project administration, Writing – review and editing; Haibin Xia, Conceptualization, Resources, Supervision, Funding acquisition, Project administration, Writing – review and editing

## Author ORCIDs

Liangliang Fu ⓘ https://orcid.org/0000-0001-8962-7739
Qin Zhao ⓘ https://orcid.org/0000-0003-4223-9522
Haibin Xia ⓘ https://orcid.org/0000-0003-2550-1146

## Ethics

All individuals provided written informed consent and this study was approved by the Ethics Committee of School & Hospital of Stomatology Wuhan University (WDKQ2024B01).
The animal experiments were conducted according to the ARRIVE guidelines 2.0. The protocol was approved by the Animal Research Ethics Committee at the School & Hospital of Stomatology, Wuhan University (No. S07922040A).

Reviewer #2 (Public review): https://doi.org/10.7554/eLife.96908.3.sa1
Author response https://doi.org/10.7554/eLife.96908.3.sa2

# Additional files

## Supplementary files

MDAR checklist

Supplementary file 1. Tables of this manuscripts.

## Data availability

Figure 1—Source data 1 is uncropped western blots with labeling for panel E. Figure 1—Source data 2 is original tiff files of western blots for panel E. Figure 7—figure supplement 2—Source data 1 is uncropped western blots with labeling for panel D. Figure 7—figure supplement 2—Source data 2 is original tiff files of western blots for panel D. Coding scripts have been provided at (https://github.com/gougou110/cd81-senescent-like copy archived at *gougou110, 2025*). Single-cell RNA-sequencing data obtained in this study are provided in NIH Gene Expression Omnibus (GSE164241, GSE152042, and GSE242714). All other data needed to evaluate the conclusions of this study are present in the paper.

The following previously published datasets were used:

| Author(s) | Year | Dataset title | Dataset URL | Database and Identifier |
|-----------|------|---------------|-------------|-------------------------|
| Yianni V, Caetano AJ, Sharpe PT | 2020 | Transcriptomic profiling of human gingiva in health and disease | https://www.ncbi.nlm.nih.gov/geo/query/acc.cgi?acc=GSE152042 | NCBI Gene Expression Omnibus, GSE152042 |

*Continued on next page*

*Continued*

| Author(s) | Year | Dataset title | Dataset URL | Database and Identifier |
|---|---|---|---|---|
| Williams DW, Moutsopoulos NM | 2021 | Single-cell atlas of human oral mucosa reveals a stromal-neutrophil axis in tissue immunity regulation | https://www.ncbi.nlm.nih.gov/geo/query/acc.cgi?acc=GSE164241 | NCBI Gene Expression Omnibus, GSE164241 |
| Neves VC, Menon Kallayil A | 2023 | Gene expression profile of the impact of Metformin on the gingiva for periodontal disease prevention | https://www.ncbi.nlm.nih.gov/geo/query/acc.cgi?acc=GSE242714 | NCBI Gene Expression Omnibus, GSE242714 |

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
