## [Editor Report · eLife Assessment]

This **valuable** study identifies a population of CD81-positive fibroblasts showing senescence signatures that can activate neutrophils through the C3/C3aR1 axis, hence contributing to the inflammatory response in periodontitis. **Solid** evidence, combining in vitro and in vivo analyses and mouse and human data, supports these findings. The revised manuscript has addressed many concerns significantly. The work would be of interest to researchers working in the senescence and oral medicine fields.

---

## [Referee Report · Reviewer #2 (Public review)]

Summary:

The authors report the discovery of a population of gingival fibroblasts displaying the expression of cellular senescence markers P21 and P16 in human periodontitis samples and a murine ligature-induced periodontitis (LIP) model. They support this finding in the murine model through bulk RNA-sequencing and show that differentially expressed genes are significantly enriched in the SenMayo cellular senescence in aging dataset. They then show that Ligature-Induced Periodontitis (LIP) mice treated with the senomorphic drug metformin display overall diminished bone damage, reduced histomorphic alterations, and a reduction in P21 and P16 immunostaining signal. To explore the cell types expressing cellular senescence markers in periodontitis, the authors make use of a combination of bioinformatic analyses on publicly available scRNA-seq data, immunostainings on patient samples and their LIP model; as well as in vitro culture of healthy human gingival fibroblasts treated with LPS. They found that fibroblasts are a cell population expressing P16 in periodontitis which are also enriched for SenMayo genes, suggesting they have a senescent phenotype. They then point to a subgroup of fibroblasts expressing CD81+ with the highest enrichment for a SASP geneset in periodontitis. They also show that treatment of LIP mice and human LPS-treated gingival fibroblasts with metformin leads to a reduction of P21 and P16-positive cells, as well as the senescence-associated beta-galactosidase (SA-beta-gal) marker. Finally, they show evidence suggesting that CD81+ senescent fibroblasts are the source of C3 complement protein, which they stipulate signals through the C3AR1 receptor present in neutrophils in periodontitis. The authors observed that both CD81+ fibroblast and C3AR1+ neutrophil populations are expanded in periodontitis, that both populations appear to be in close contact, and that treatment with metformin reduced both C3 and the neutrophil marker MPO in their mouse LIP model.

After a round of revision, the authors have made significant improvements to their manuscript, such as improving the quality of the data/evidence and also included new data from experiments using a well-known senolytic and the senomorphic metformin, which all together provide a solid support to their main claims.

Strengths:

The study implements several different techniques and tools on human samples, mouse models, fibroblast cultures, and publicly available data to support their conclusions. In summary, they provide solid evidence showing that in the context of periodontitis, there is an expansion of cells expressing senescence markers P21, and P16, as well as members of the SASP, and that this includes CD81+ fibroblasts.

Weaknesses:

The fact that in this study the periodontitis samples belonged to patients with a significantly higher median age (all older than 50 years of age) and the healthy samples belonged to young adults (all younger than 35 years of age), raises the need for caution in interpretation due to a possible effect of aging in the accumulation of CD81+ senescent fibroblasts. However, the recruitment of similar age groups in this case is of course difficult due to the higher prevalence of periodontitis in older adults. In this regard it is important to note that the authors still support their findings using a mouse ligature model. Similar studies comparing healthy and periodontitic patients from similar age groups will be of great importance in the future.

---

## [Author Response]

The following is the authors’ response to the original reviews

**Recommendations for the authors:**

**Reviewing Editor (Recommendations For The Authors):**
There are four main areas that need further clarification:(1) Further and more complete assessment of senescence and the fibroblasts must be done to support the claims.

We sincerely appreciate the Reviewing Editor's valuable suggestion regarding the addition of cellular senescence detection markers. In the revised manuscript, we have incorporated additional detection markers for cellular senescence, such as H3K9me3 and SA-β-gal staining, in healthy and periodontitis gingival samples to further validate our findings (Figure 1A, B in revised manuscripts).

(2) Confusion between ageing and senescence throughout the manuscript.

We fully understand the concerns raised by the Reviewing Editor and reviewers regarding the confusion between the concepts of ageing and senescence in the manuscript. Cellular senescence is a manifestation of ageing at the cellular level. In the revised manuscript, we have given priority to the term ‘senescence’ to describe the cell condition instead of ‘aging’.

(3) The lipid metabolism mechanistic claims are very speculative and largely unsupported by experimental data.

We greatly appreciate the Reviewing Editor and reviewers for pointing out the incorrect statements regarding the role of lipid metabolism in regulating cellular senescence. Since the mechanism by which cellular metabolism regulates cellular senescence is not the core focus of this manuscript, we have moved the results of the metabolic analysis from the sc-RNA sequencing data to the figure supplement (Figure 4-figure supplement 1) and revised the related statements in the revised manuscript (Page 7-8, Line 186-194).

(4) Concerns about the use of Metformin as a senotherapy vs other pleiotropic effects in periodontitis and the suggestion of using an alternative Senolytic drug (Bcl2 inhibitors, etc.).

We fully understand the concerns of the Reviewing Editor and reviewers regarding metformin as an anti-aging therapy. In the revised manuscript, we have included additional experiments using other senolytic drugs ABT-263, a Bcl2 inhibitor, in the ligature-induced periodontitis mouse model. The corresponding results could be found in the Figure 6. and Page 9-10, Line 248-264 in the revised manuscripts.

**Reviewer #1 (Recommendations For The Authors):**
While most of the experiments are elegantly designed and the procedures well conducted there are several critical weaknesses that temper my enthusiasm for this solid and timely work. Considering my main points, I would recommend the following:(1) Potentiate the senescent assessment in vitro and, most importantly, in vivo. E.g. SABgal with fresh tissue, other senescent biomarkers like SAHFs (HP1g or H3K9me3), etc.

We sincerely appreciate the reviewers' suggestion to potentiate the assessment of cellular senescence. In the revised manuscript, we performed SA-β-gal staining on fresh frozen samples, revealing a significantly higher number of SA-β-gal positive cells in the gingival tissue of periodontitis, particularly in the lamina propria, while few SA-β-gal positive cells were observed in healthy gingival tissue (Figure. 1A). Additionally, we assessed the protein level changes of H3K9me3, a marker of senescence-associated heterochromatin foci (SAHF), in gingival tissues from healthy individuals and periodontitis patients. The results showed a notable increase in the number of H3K9me3 positive cells in periodontitis tissues, approximately double that found in healthy gingiva (Figure. 1B). This trend aligns with our previous findings of elevated p16 and p21 levels. Collectively, these results further confirm that periodontitis gingival tissues contain a greater number of senescent cells compared to healthy gingiva.

(2) Claims on disturbances in lipid metabolism as a driver of CD81+ fibroblast senescence require appropriate functional/mechanistic validations and experiments of metabolism rewiring.

We sincerely appreciate the reviewers' suggestion for more experimental evidence regarding the role of lipid metabolism in driving CD81+ fibroblast senescence. The influence and mechanisms of lipid metabolism on cellular senescence is a complex and important scientific issue, and it is not the central focus of this manuscript. Therefore, to avoid causing confusion for the reviewers and readers, we have removed the metabolism analysis in the Figure 4-figure supplement 1 and revised the presentation of the relevant results in the revised manuscript to ensure a more rigorous interpretation of our findings (Page 7-8, Line 186-194).

(3) Do LPS-stimulated HGFS implementing the senescent programme secrete C3? Detection of complement C3 at the protein level (e.g. by ELISA) would reinforce the proposed mechanism.

This is indeed a very interesting question. In response to the reviewers' suggestion, we measured the levels of C3 protein secreted by human gingival fibroblasts induced by Pg-LPS, which is one of the markers of the senescence-associated secretory phenotype (SASP). The results indicated that, compared to untreated fibroblasts, those induced by Pg-LPS exhibited significantly higher levels of C3 secretion, approximately 1.5 times that of the control group (Figure. 5G). Additionally, we also found that primary gingival fibroblasts derived from periodontitis tissues secreted more complement C3 compared to those derived from healthy tissues (Figure. 5F). These findings suggest that the increased secretion of complement C3 by gingival fibroblasts in periodontitis tissues may be related to Pg-LPS-induced cellular senescence.

(4) The mechanism of Metformin to impair senescence and/or the SASP is not fully validated and Metformin can produce other pleiotropic effects. A key experiment (including therapeutic implications) is using a senolytic drug (e.g. Navitoclax) to causally connect the eradication of senescent CD81+ fibroblasts with the recruitment of neutrophils. If the hypothesis of the authors is correct this approach should result in reduced levels of gingival CD81 and C3 positivity, prevention of neutrophils infiltration (reduced MPO positivity), and ameliorate bone damage in ligationinduced periodontitis murine models.

We fully understand the reviewers' concerns regarding the role of metformin in alleviating cellular senescence and the possibility of it acting through non-senescent pathways. To clarify the role of cellular senescence in the recruitment of neutrophils by CD81+ fibroblasts through C3 in periodontitis, we treated a ligature-induced periodontitis mouse model with ABT-263, also known as Navitoclax. The results showed that after ABT-263 treatment, the number of p16-positive or H3K9me3-positive senescent cells in the periodontitis mice significantly decreased. Additionally, we observed reductions in the quantities of CD81+ fibroblasts, C3 protein levels, neutrophil infiltration, and osteoclasts to varying degrees in the LIP model after ABT263 treatment (Figure. 6). These results further support our hypothesis that the eradication of senescent CD81+ fibroblasts could reduce neutrophil infiltration and alveolar bone resorption.

(5) Have the authors considered using any of the available C3/C3aR inhibitors to validate the involvement of neutrophils and the inflammatory response in periodontitis? A C3/C3aR inhibitor would be an elegant treatment group in parallel with the senolytic approach.

Thank you very much for the reviewers' suggestion to investigate neutrophil infiltration and inflammatory responses after treating periodontitis with C3/C3aR inhibitors. In a clinical study by Hasturk et al. in 2021 (Reference 1), it was found that using the C3 inhibitor AMY-101 effectively alleviated gingival inflammation levels in periodontitis patients. This was reflected in significant decreases in clinical indicators such as the modified gingival index and bleeding on probing, as well as a marked reduction in inflammatory tissue destruction markers, including MMP-8 and MMP-9. In addition, Tomoki Maekawa et al. (Reference 2) demonstrated that a peptide inhibitor of complement C3 effectively reduced inflammation levels and the extent of bone resorption in periodontitis. Moreover, research by Guglietta et al. (Reference 3) clarified that the C3 complement promotes neutrophil recruitment and the formation of neutrophil extracellular traps (NETs) via C3aR. And neutrophil extracellular traps are considered key pathological factors in causing sustained chronic inflammation in periodontitis (References 4 and 5). In summary, existing studies have clearly indicated that C3/C3aR inhibitors likely reduce neutrophil recruitment and inflammation in periodontitis.

Reference

(1) Hasturk, H., Hajishengallis, G., Forsyth Institute Center for Clinical and Translational Research staff, Lambris, J. D., Mastellos, D. C., & Yancopoulou, D. (2021). Phase IIa clinical trial of complement C3 inhibitor AMY-101 in adults with periodontal inflammation. The Journal of clinical investigation, 131(23), e152973.

(2) Maekawa, T., Briones, R. A., Resuello, R. R., Tuplano, J. V., Hajishengallis, E., Kajikawa, T., Koutsogiannaki, S., Garcia, C. A., Ricklin, D., Lambris, J. D., & Hajishengallis, G. (2016). Inhibition of pre-existing natural periodontitis in non-human primates by a locally administered peptide inhibitor of complement C3. Journal of clinical periodontology, 43(3), 238–249.

(3) Guglietta, S., Chiavelli, A., Zagato, E., Krieg, C., Gandini, S., Ravenda, P. S., Bazolli, B., Lu, B., Penna, G., & Rescigno, M. (2016). Coagulation induced by C3aR-dependent NETosis drives protumorigenic neutrophils during small intestinal tumorigenesis. Nature communications, 7, 11037.

(4) Kim, T. S., Silva, L. M., Theofilou, V. I., Greenwell-Wild, T., Li, L., Williams, D. W., Ikeuchi, T., Brenchley, L., NIDCD/NIDCR Genomics and Computational Biology Core, Bugge, T. H., Diaz, P. I., Kaplan, M. J., Carmona-Rivera, C., & Moutsopoulos, N. M. (2023). Neutrophil extracellular traps and extracellular histones potentiate IL-17 inflammation in periodontitis. The Journal of experimental medicine, 220(9), e20221751.

(5) Silva, L. M., Doyle, A. D., Greenwell-Wild, T., Dutzan, N., Tran, C. L., Abusleme, L., Juang, L. J., Leung, J., Chun, E. M., Lum, A. G., Agler, C. S., Zuazo, C. E., Sibree, M., Jani, P., Kram, V., 6 Martin, D., Moss, K., Lionakis, M. S., Castellino, F. J., Kastrup, C. J., … Moutsopoulos, N. M. (2021). Fibrin is a critical regulator of neutrophil effector function at the oral mucosal barrier. Science (New York, N.Y.), 374(6575), eabl5450.

Other comments(1) Figure 1. The authors report upregulation of the aging pathway in bulk RNAseq analyses. What about the upregulation of senescence-related pathways and differential expression of SASP-related genes in this experiment?

Thanks for this interesting question. Through further analysis of the bulk RNA sequencing results of gingival tissues from LIP mice model, we found significant alterations in multiple senescence-associated secretory phenotype (SASP) genes and several cellular senescencerelated pathways. SASP genes, such as Icam1, Mmp3, Nos3, Igfbp7, Igfbp4, Mmp14, Timp1, Ngf, Il6, Areg, and Vegfa, were markedly upregulated in the periodontitis samples of ligature-induced mice (Figure 1-figure supplement 2A). Moreover, we observed a significant reduction in oxidative phosphorylation levels and the tricarboxylic acid (TCA) cycle in the periodontitis group, suggesting that the occurrence of cellular senescence may be related to mitochondrial dysfunction (Figure 1figure supplement 2B and C.).

Additionally, we noted the activation of the PI3K-AKT and MAPK pathways in LIP model (Figure 1-figure supplement 2D and E), both of which can induce cellular senescence by activating the tumor suppressor pathway TP53/CDKN1A, leading to cell cycle arrest (References 1, 2). Furthermore, the NF-κB signaling pathway was also significantly enriched in LIP model (Figure 1-figure supplement 2F), which is closely associated with the secretion of SASP factors (Reference 3).

In summary, our bulk RNA sequencing results suggest enrichment of cellular senescencerelated pathways in the periodontitis group, including mitochondrial metabolic dysregulation, senescence-related pathways, and alterations in the SASP. Related results were added into Page 56 of the revised manuscripts.

Reference

(1) Tang Q, Markby GR, MacNair AJ, Tang K, Tkacz M, Parys M, Phadwal K, MacRae VE, Corcoran BM. TGF-β-induced PI3K/AKT/mTOR pathway controls myofibroblast differentiation and secretory phenotype of valvular interstitial cells through the modulation of cellular senescence in a naturally occurring in vitro canine model of myxomatous mitral valve disease. Cell Prolif. 2023 Jun;56(6):e13435. doi: 10.1111/cpr.13435.

(2) Sayegh S, Fantecelle CH, Laphanuwat P, Subramanian P, Rustin MHA, Gomes DCO, Akbar AN, Chambers ES. Vitamin D3 inhibits p38 MAPK and senescence-associated inflammatory mediator secretion by senescent fibroblasts that impacts immune responses during ageing. Aging Cell. 2024 Apr;23(4):e14093.

(3) Raynard C, Ma X, Huna A, Tessier N, Massemin A, Zhu K, Flaman JM, Moulin F, Goehrig D, Medard JJ, Vindrieux D, Treilleux I, Hernandez-Vargas H, Ducreux S, Martin N, Bernard D. NF-κB-dependent secretome of senescent cells can trigger neuroendocrine transdifferentiation of breast cancer cells. Aging Cell. 2022 Jul;21(7):e13632.

(2) I wonder whether the authors could clarify how the semi quantifications for p21, p16, Masson's trichrome, C3, or MPO were done in Figures 1, 2, and 6.

Thank you very much for the reviewer's suggestion. We have added the semi-quantitative methods for p21, p16, Masson's trichrome, C3, and MPO in the Methods section. Specifically, for semi-quantification of protein expressions, the mean optical density (MOD) of positive stains for p21, p16, and C3 was measured using the ImageJ2 software (version 2.14.0, National Institutes of Health, Bethesda, MD). The number of MPO-positive cells and collagen volume fractions (stained blue) for individual sections were also measured using the ImageJ2 software. (Page 19, Line 537-541 in the revised manuscripts).

(3) Figure 2. It is unclear whether N=6 refers to 6 mice, maxilla, or fields per group.

Thank you very much for the reviewer's question. To avoid any misunderstandings for the reviewer and readers, we have added a definition of the sample size in the description of the micro-CT analysis method. Specifically, in the micro-CT quantitative analysis, the sample size n for each group consists of 6 mice, with the average value of the BV/TV of the bilateral maxillary alveolar bone taken as one sample for statistical analysis (Page 17-18, Line 488-490 in the revised manuscripts).

(4) Figure 4K. Please provide separated staining for p16, VIM, and CD81, and not only the Merge. It is difficult to identify the triple-positive cells. Also, the arrows are difficult to observe.

Thank you very much for the reviewer's suggestion. In the revised manuscript, we have included separated staining for p16, VIM, and CD81, and the triple-positive cells are indicated with white arrows (Figure 5-figure supplement 1).

(5) Overall, improve the magnifications in the IF experiments and show where the magnified areas come from.

Thank you very much for the reviewer's suggestion. We have enlarged the fluorescence result images.

(6) Refer to the original datasets of the scRNAseq results in figure legends.

Thank you very much for the reviewer's suggestion. We have indicated the source of the raw single-cell sequencing data in the figure legend.

(7) Check English grammar and writing.

Thank you for the reviewer's suggestion. We checked the grammar and writing in the revised manuscript assisted by a native English speaker and AI tools like Chat-GPT.

**Reviewer #2 (Recommendations For The Authors):**
(1) When the authors refer to accelerated aging and/or senescence, they are doing so in comparison to what?

Thank you for the reviewer's question, which allows me to further clarify the concepts of accelerated aging and/or senescence. In sections 2.1 and Figure 1 of this manuscript, we referred to accelerated aging and/or senescence. This indicates that the gingival tissues of periodontitis patients exhibit a higher number of senescent cells and elevated levels of senescence-related markers compared to healthy gingival tissues. In the title of this manuscript, we describe CD81+ fibroblasts as a unique subpopulation with accelerated cellular senescence. This means that CD81+ fibroblasts display higher expression levels of senescence-related genes, cell cycle inhibitor p16, and SASP factors compared to other fibroblast subpopulations. To avoid any misunderstanding, we have deleted the text ‘accelerated senescence’ in the revised manuscripts.

(2) In general, the main text does not describe the results using exact and reproducible terminology. Phrases like "X was most active", "a significant increase was observed", "the highest proportion was", and "the level of aging increased" should be supported by adding quantification details and by detailing what these comparisons are made to, to improve the reproducibility of the results.

Thank you for the reviewer's suggestion. To improve the reproducibility of the results, we have added quantification details in the results section and clarified what comparisons are being made through the whole manuscript.

(3) In some sections of the main text and figure legends, it is not entirely clear which sequencing experiments were conducted by the authors, which analyses were conducted by the authors on publicly available sequencing data, and which analyses were conducted on their mouse sequencing data.

Thank you for the valuable feedback from the reviewer. To further clarify the source of the sequencing data, we have clearly indicated the data source in both the results section and the figure legends.

(4) In Figure 3H, the images showing SA-beta-gal staining on LPS-treated fibroblasts do not show convincingly the difference between treatments that are represented in the graph.

Thank you for the reviewer's suggestion. To further clearly show the differences between treatments, we have enlarged the partial image of SA-β-gal staining shown in Figure 2-figure supplement 2 of the revised manuscripts.

(5) The choice of colors for Figure 4K is far from ideal as it is very difficult to tell apart red from purple channels and thus to visualize triple positive cells. A different LUT should be chosen, and separate individual channels should be shown to clearly identify triple-positive cells from others. Arrows also do not currently point at triple-positive cells.

Thank you for the reviewer's suggestion. In the revised manuscript, we have included separated staining for p16, VIM, and CD81, and the triple-positive cells are marked with white arrows shown in Figure 5-figure supplement 1 of the revised manuscripts.

(6) The authors state that treatment with metformin "alleviated.... inflammatory cell infiltration (Figure 2C), and collagen degradation (Figure 2D) as observed through H&E and Masson staining." However, I cannot find a description of how the "relative fraction of collagen" in Figure 2Gc was calculated and how the H&E image they provide shows evidence of a reduction in inflammatory cells at that magnification.

Thank you for the reviewer's suggestion. In the revised manuscript, we have added details in the methods section regarding the calculation of the "relative fraction of collagen" (Page 19, Line 539-541). Specifically, the collagen volume fractions (stained blue) for individual sections were measured using ImageJ2 software. Additionally, we have marked the infiltrating inflammatory cells in the gingiva in the H&E images with black arrows shown in Figure 7-figure supplement 1B of the revised manuscripts.

(7) It appears that the in vivo experiment for metformin treatment was conducted with 6 animals per group, but this is not clear in the figures, main text, and methods.

Thank you for the reviewer's suggestion. In the revised manuscript, we have included the number of mice in each group for the in vivo experiments, specifying that there are 6 mice per group in the figures, main text, and methods sections.

(8) The methodology described for the bulk RNA-sequencing experiment in mice should describe the sequencing library characteristics and some reference to quality control thresholds that were implemented (mapped and aligned reads, sequencing depth and coverage, etc.).

In the bulk RNA-sequencing experiment, the sequencing library characteristics and quality control thresholds were listed as follows:

Sequencing Library Characteristics: We utilized the Illumina TruSeq RNA Library Construction Kit, generating libraries with an insert fragment length of approximately 400-500 bp.

Quality Control Standards include the following:

Alignment and Mapping Rates: The read data for all samples underwent preliminary quality control using FastQC (v0.11.9) and were aligned using HISAT2 (v2.2.1). The average mapping rate for each sample was over 90%.

Sequencing Depth and Coverage: Each sample had a sequencing depth of 30M-40M paired reads to ensure sufficient transcript coverage. Detailed alignment statistics have been provided in the supplementary materials.

Other Quality Control Measures: During the analysis, we also utilized RSeQC (v3.0.1) to evaluate the transcript coverage and GC bias of the sequencing data.

The corresponding method description and reference were added in the Page 19-20, Line 546-558 of the revised manuscripts.

(9) Patients with periodontitis are labeled as diagnosed with "chronic periodontitis". I would like to know how the authors defined this chronic state of the disease in their inclusion criteria.

Thank you very much for the reviewer’s question, which gives us the opportunity to further clarify the definition and diagnosis of chronic periodontitis. The diagnostic criteria for patients with chronic periodontitis in this study are based on the 1999 International Workshop for a Classification of Periodontal Diseases and Conditions (Reference 1). Chronic periodontitis is a type of periodontal disease distinct from aggressive periodontitis, and it is not diagnosed based on the rate of disease progression. Clinically, the diagnosis of chronic periodontitis is primarily based on clinical attachment loss (CAL) ≥ 4 mm or probing depth (PD) ≥ 5 mm as one of the criteria for diagnosis.

Reference

(1) Armitage G. C. (2000). Development of a classification system for periodontal diseases and conditions. Northwest dentistry, 79(6), 31–35.

(10) There is no detail about the age and sex of the donors for the healthy gingival fibroblast experiments. Are they some of the patients mentioned in Supplementary Table 1? Please clarify the source and number of independent primary cultures.

Thank you very much to the reviewer for allowing us to further clarify the source and number of independent primary cultures. In the cell experiments, we used gingival fibroblasts derived from gingival tissue of two healthy volunteers and two patients with periodontitis as experimental subjects. This information has been listed in the Supplementary Table 1.

(11) Can the authors explain why their age inclusion criteria were different for the healthy and periodontitis groups according to their methods (healthy 18-50 years old: periodontitis 18-35 years old?)

Thank you very much to the reviewer for pointing this out. We noticed that there was an error in the age range indicated for the healthy and periodontitis groups in the inclusion criteria. Based on the original inclusion criteria information, we have corrected the age range of the included population. 18-65 years old individuals were included into the both healthy and periodontitis groups. (Page 14-15, Line 396-404 in the revised manuscripts)

(12) The methodology for inclusion is confusing and does not reflect the actual information of the recruited patients and samples thus analyzed. In the text, the healthy group appears to have included 8 young adult individuals and 8 middle-aged individuals. However, the list of recruited patients shows all healthy patients were in the young adult range (below 35 years of age) while all chronic periodontitis patients were middle-aged (above 50 years of age). Please clarify.

Thank you very much to the reviewer for pointing out the issues in the article. This study included 8 healthy periodontal patients and 8 patients with periodontitis (Page 14, Line 396-398 and Supplementary Table 1 in the revised manuscripts). Since periodontitis has a higher prevalence in middle-aged and elderly populations, the periodontitis samples included in this study were mostly from this demographic. In contrast, the healthy gingival samples were sourced from patients undergoing wisdom tooth extraction, which primarily involves younger individuals. Therefore, due to the limited sample size, we could not enforce strict age matching. To address this, we repeated the relevant experiments in more consistent mouse models, which confirmed the increase in senescent cells in periodontal tissues (Figure 1D in the revised manuscripts). In summary, although the clinical samples were limited, the experimental results from the mouse models still support our conclusions.

(13) The number of biological replicates for each group used in the bulk RNA-sequencing experiment is unclear. The methods state:" For those with biological duplication, we used DESeq2 [8] (version: 1.34.0) to screen differentially expressed gene sets between two biological conditions; for those without biological duplication, we used edgeR". Please clarify the number of mouse samples sequenced and the description of the groups.

Thank you very much to the reviewer for pointing out the errors in the article. In the transcriptome sequencing, we collected gingival tissues from 3 healthy mice and gingival tissues from 3 ligature-induced periodontitis mice. Therefore, we used the DESeq2 (version: 1.34.0) method to filter for differentially expressed genes. The corresponding descriptions were revised in Page 20, Line 554-555 in the revised manuscripts.

(14) Cluster group labels are misaligned in Figure 4C.

Thank you very much for the reviewer's suggestion. The cluster group labels in Figure 3C of the revised manuscripts have been aligned.

**Reviewer #3 (Recommendations For The Authors):**
Major Comments for the Authors:(1) I do not find the immunohistochemical staining of p16 and p21 shown in Figures 2E and F to be particularly compelling. Especially as other stains of these markers used later in the manuscript are of higher quality (i.e. Figures 3F and G). Can this staining be improved to better reflect the quantifications in Figure 2G?

Thank you very much for the reviewer's suggestion. In the revised manuscript, we have provided more representative images in Figure 7C in the revised manuscripts to reflect the effect of metformin treatment on the number of p16-positive cells in periodontitis. In Figure 7-figure supplement 1D of the revised manuscripts, we have marked p21-positive cells with black arrows to help readers better identify the p21-positive cells. Additionally, we have also assessed the H3K9me3 marker, which is more specific, and the results similarly indicate that metformin treatment can alleviate the formation of senescent cells in periodontitis (Figure 7-figure supplement 1E of the revised manuscript).

(2) On line 140, Supplementary Figure 2C, D is quoted to show "...an increase in senescence characteristics of fibroblasts with the severity of periodontitis." This figure panel does not appear to support this statement. Please revise.

Thank you very much for pointing out the errors in the manuscript. In the revised version, we have corrected this part of the description and added that “The results showed a decline in fibroblast proportion along with increasing disease severity (Figure 2-figure supplement 1C and D)” (Page 6, Line 153-154 of the revised manuscript)

(3) I do not find the Western Blot experiment in Figure 4L to be particularly convincing. The text states that p21, p16, and CD81 increase in a context-dependent manner upon LPS stimulation, which doesn't appear to be very evident. I recommend repeating this experiment and showing both a representative blot alongside a blot density quantification where the bars have the error shown between experiments.

Thank you very much for the reviewer’s suggestion regarding this result. During subsequent repeated experiments, we found that the result was not reproducible, and we have removed the related results.

(4) The results state that metabolic profiling of senescent fibroblasts shows an increase in the biosynthesis of Linoleic acid, linolenic acid, arachidonic acid, and steroid. However, in Figure 5B only arachidonic acid and steroid biosynthesis appear to be elevated in CD81+ Fibroblasts, while Linoleic and linolenic acid appear to be decreased. Can the authors comment on this discrepancy? Moreover, in Figure 5C steroid biosynthesis is unchanged between healthy and periodontitis samples, contrary to the claimed increased trend in the results text. Please revise this section. Also, in Figures 5 B and C some of the terms are highlighted in a red or blue box. This is not discussed in the figure legend. Could the significance of this be explained or could these highlights be removed from the figure?

Thank you very much for the reviewer’s correction regarding the errors in the manuscript. In the Page 7-8, Line 186-194 of the revised manuscripts, “Pathways related to fatty acid biosynthesis, arachidonic acid metabolism, and steroid biosynthesis were significantly upregulated in CD81+ fibroblasts (Figure 4-figure supplement 1A)” was re-wrote. Moreover, we have removed the results from Figure 5C, and the highlights in Figures 5B and C of the previous manuscripts. Since the mechanism by which cellular metabolism regulates cellular senescence is not the core focus of this manuscript, we have moved the results of the metabolic analysis from the sc-RNA sequencing data to the figure supplement (Figure 4-figure supplement 1) and revised the related statements in the revised manuscript (Page 7-8, Line 186-194).

(5) The authors state that arachidonic acid can be converted to prostaglandins and leukotrienes through COXs (which are expressed in their CD81+ Fibroblasts), accentuating inflammatory responses. Have the authors profiled for the expression of prostaglandins and leukotrienes in their CD81+ Fibroblasts or between healthy and periodontitis samples? Such data would be a great inclusion in the manuscript.

Thank you very much for the reviewer’s suggestion. Our results indicated that CD81+ gingival fibroblasts expressed higher levels of PTGS1 and PTGS2 compared to other fibroblast subpopulations. These genes encode proteins that are COX-1 and COX-2, which are key enzymes in prostaglandin biosynthesis (Figure 4-figure supplement 1 of the revised manuscript). Additionally, previous studies have reported high levels of prostaglandins and leukotrienes in periodontal tissues, and these pro-inflammatory mediators contribute to tissue destruction in periodontitis (Reference 1 and 2).

Reference

(1) Van Dyke, T. E., & Serhan, C. N. (2003). Resolution of inflammation: a new paradigm for the pathogenesis of periodontal diseases. Journal of dental research, 82(2), 82–90.

(2) Hikiji, H., Takato, T., Shimizu, T., & Ishii, S. (2008). The roles of prostanoids, leukotrienes, and platelet-activating factor in bone metabolism and disease. Progress in lipid research, 47(2), 107–126.

(6) Lines 199 and 200 state "...the cellular senescence of CD81+ fibroblasts could be attributed to disturbances in lipid metabolism". While altered lipid metabolic profiles are shown in Figure 5 to correlate with senescent fibroblasts/periodontitis tissue, no evidence is shown to suggest that they are the driver or cause of fibroblast senescence. Could this sentence be amended to better reflect the conclusions that can be drawn from the data presented?

Thank you very much for the reviewer’s suggestion. We have revised the related statements and believed that “lipid metabolism might play a role in cellular senescence of the gingival fibroblasts” in the Page 7, Line 189 of the revised manuscripts.

Minor Comments for the Authors:(1) There are some sentences without references that I feel would warrant referencing: - Line 112 - "Metformin, an anti-aging drug has shown potential in inhibiting cell senescence in various disease models (REFERENCE)."

Thank you for the reviewer's suggestion. We have included the relevant references in the Page10, Line 267-271 of the revised manuscripts.

Reference

(1) Soukas, A. A., Hao, H., & Wu, L. (2019). Metformin as Anti-Aging Therapy: Is It for Everyone?. Trends in endocrinology and metabolism: TEM, 30(10), 745–755.

(2) Kodali, M., Attaluri, S., Madhu, L. N., Shuai, B., Upadhya, R., Gonzalez, J. J., Rao, X., & Shetty, A. K. (2021). Metformin treatment in late middle age improves cognitive function with alleviation of microglial activation and enhancement of autophagy in the hippocampus. Aging cell, 20(2), e13277.

- Line 210 - "Previous studies have demonstrated the importance of sustained neutrophil infiltration in the progression of periodontitis (REFERENCE)."

Thank you for the reviewer's suggestion. We have included the relevant references in the Page 8, Line 211-214 of the revised manuscripts.

Reference

(1) Song, J., Zhang, Y., Bai, Y., Sun, X., Lu, Y., Guo, Y., He, Y., Gao, M., Chi, X., Heng, B. C., Zhang, X., Li, W., Xu, M., Wei, Y., You, F., Zhang, X., Lu, D., & Deng, X. (2023). The Deubiquitinase OTUD1 Suppresses Secretory Neutrophil Polarization And Ameliorates Immunopathology of Periodontitis. Advanced science (Weinheim, Baden-Wurttemberg, Germany), 10(30), e2303207.

(2) Kim, T. S., Silva, L. M., Theofilou, V. I., Greenwell-Wild, T., Li, L., Williams, D. W., Ikeuchi, T., Brenchley, L., NIDCD/NIDCR Genomics and Computational Biology Core, Bugge, T. H., Diaz, P. I., Kaplan, M. J., Carmona-Rivera, C., & Moutsopoulos, N. M. (2023). Neutrophil extracellular traps and extracellular histones potentiate IL-17 inflammation in periodontitis. The Journal of experimental medicine, 220(9), e20221751.

(3) Ando, Y., Tsukasaki, M., Huynh, N. C., Zang, S., Yan, M., Muro, R., Nakamura, K., Komagamine, M., Komatsu, N., Okamoto, K., Nakano, K., Okamura, T., Yamaguchi, A., Ishihara, K., & Takayanagi, H. (2024). The neutrophil-osteogenic cell axis promotes bone destruction in periodontitis. International journal of oral science, 16(1), 18.

(2) To improve the quality of several of the authors' claims I would recommend some further quantification of their experimental analyses. Namely:- Figures 3 F and G- Figures 4 I, J and K- Figures 6 F and G- Supplementary Figures 4 A, B, and C

Thank you for the reviewer's suggestion. We have supplemented the quantitative analysis results for some images based on the reviewer's recommendations, specifically in Figure. 2G, Figure. 3G, Figure 5-figure supplement 1A, B, Figure 5-figure supplement 2A and Figure 7figure supplement 3A-D in the revised manuscripts.

(3) Figure 1L has missing x-axis annotation.

Thank you for the reminder from the reviewer. The X-axis label has been added in Figure 1-figure supplement 1D for the GO term annotation.

(4) Line 117 is missing a reference for the experimental schematic shown in Figure 2A.

Thank you for the reminder from the reviewer. The experimental schematic shown in Figure 7A has been referenced in Page 10, Line 275-277.

(5) The "BV/TV ratio" and "CEJ-ABC distance" should be briefly explained in the results test (Lines 118 and 119).

Thank you for the reviewer's suggestion. We have added the explanation of "BV/TV ratio" and "CEJ-ABC distance." In Page 10-11, Line 279-281 in the revised manuscripts.

(6) Figure 2 could be improved by having some annotation for the anatomical regions shown.

Thank you for the reviewer’s valuable suggestion. We have labeled the relevant anatomical structures to enhance clarity in Figure 7 in the revised manuscripts.

(7) The positive signal for p16 and p21 is difficult to interpret in Figure 2. Could the clarity of this be improved either by using more evident images or annotation with arrowheads indicating positive cells?

Thank you for the reviewer's suggestion. In the revised manuscript, we have provided more representative images in Figure. 7C in the revised manuscripts to reflect the effect of metformin treatment on the number of p16-positive cells in periodontitis. In Figure 7-figure supplement 1D of the revised manuscripts, we have marked p21-positive cells with black arrows to help readers better identify the p21-positive cells. Additionally, we have also assessed the H3K9me3 marker, which is more specific, and the results similarly indicate that metformin treatment can alleviate the formation of senescent cells in periodontitis (Figure 7-figure supplement 1E of the revised manuscript).

(8) Figure 2Gc, d, and e are not mentioned in the results text. Please include references to these panels at the appropriate points.

Thank you for the reminder. In the revised manuscripts, Figures 2G c, d, and e in the previous manuscripts have been mentioned in the text in the Page 11, Line 284-289 of the revised manuscript.

(9) Scale bars are missing in Supplementary Figure 2E.

Thank you for the suggestion. The scale bar has been added in the Figure 7-figure supplement 2B in the revised manuscripts.

(10) The order of the figure panels is not always mentioned in the order they are referred to in the text. For example, Figure 3 is presented in the order of A, B, D then C. Could this be changed to reflect the order in the results text?

Thank you for the feedback. We have renumbered the figures according to the order mentioned in the original manuscript (Page 6, Line 146-149, Figure 2 in the revised manuscripts).

(11) To improve reader clarity it would be good to briefly introduce the gene expression datasets analysed, such as GSE152042. I.e. what the experimental condition is from which it is derived.

Thank you for the suggestion. We have included a brief description of the information and sources of the samples from GSE152042 in Page 6, Line 140-142 of the revised manuscripts.

(12) To improve reader clarity I would recommend signifying clearly in the figure if the data shown is from mouse or human samples. For example in Figure 3F and G.

Thank you for the suggestion. We have moved all the results from the mouse experiments to the figures supplement (Figure 5-figure supplement 1 and 2 in the revised manuscripts).

(13) The images shown in Figure 3H for SA-beta-Gal do not seem very convincing. Could this be improved?

Thank you for the suggestion. To further illustrate the differences in SA-beta-Gal results between the groups, we have provided images at higher magnification in the Figure 2-figure supplement 2 of the revised manuscripts.

(14) Supplementary Figure 2E would benefit from small experimental schematics that would allow the reader to appreciate the timings of the treatment for this experiment.

Thank you for the suggestion. We have added a schematic diagram in Figure 7-figure supplement 2A of the revised manuscripts to illustrate the LPS treatment, metformin treatment, and the timing of the assessments.

(15) Figure 4K would benefit from showing the merged image and single channels of each of the stains to better assess the degree of colocalisation.

Thank you for the suggestion. We have included each individual fluorescence channel in Figure 5-figure supplement 1C of the revised manuscripts.

(16) The writing on the X-axis of Figure 6B is almost illegible to me, although this may just be a compression artefact. This makes the interpretation of the data quite difficult. Also, for Figures 6 B and C, the meaning of the (H) and (P) annotations should be clear on either the figure or figure legend. I surmise that they represent "Healthy" and "Periodontic" samples respectively.

Thank you for the suggestion. In the revised manuscript, we have enlarged Figure 6B in the previous manuscripts to better display the X-axis as shown in the Figure 5B of the revised manuscripts. Additionally, we have fully labeled "Healthy" and "Periodontitis" in Figure 5C of the revised manuscripts.

(17) MPO-positive cells are introduced on line 216, however, no explanation is provided for what population or state the expression of this protein marks. I surmise the authors are using it to detect Neutrophil populations. If so, could the authors briefly state this the first time it is used?

Thank you for the suggestion. In the revised manuscript, we have added an introduction to MPO. MPO, or myeloperoxidase, is considered one of the markers for neutrophils. (Page 9, Line 240-242 of the revised manuscripts)

(18) Supplementary Figure 3D does not appear to be mentioned or discussed in the results text.

Thank you for the reminder. We have referenced Supplementary Figure 3D in the previous manuscripts in Page 9, Line 240-242 shown as Figure 5-figure supplement 2C of the revised manuscript.

(19) Figure 6E showing increased C3 expression in periodontic samples is not very convincing and differences in expression are not evident. Can the authors provide an image that more convincingly matches their quantification?

Thank you for the suggestion. In the revised manuscript, we have provided more representative images shown in Figure 5E of the revised manuscript.

(20) Figure 6I shows the expression of CD81 and SOD2 in healthy and periodontic tissue. The associated results texts (Lines 220 to 223) discuss the spatial coincidence of CD81 and MPO. Can the authors address this discrepancy in either the results text or the figure panel? Moreover, can Figure 6H and I be annotated to show the location of the gingival lamina propria to improve clarity?

Thank you for the reminder. We have revised the relevant statements in the text: "Interestingly, spatial transcriptomic analysis of gingival tissue revealed that the regions expressing CD81 and SOD2, a neutrophil marker, in periodontitis overlapped in the gingival lamina propria, showing a high spatial correlation" in Page 9, Line 223-226 of the revised manuscripts. Additionally, we have labeled the gingival lamina propria (LP) in Figure 5H of the revised manuscripts.

(21) I am confused about the purpose of Supplementary Figure 3E and what evidence it provides. Can the authors comment on this?

Thank you for the reminder. To avoid any potential misunderstanding by readers, we have deleted Supplementary Figure 3 image in the revised manuscripts